# α-Helical peptidic scaffolds to target α-synuclein toxic species with nanomolar affinity

Jaime Santos [1], Pablo Gracia[2], Susanna Navarro [1], Samuel Peña-Díaz[1], Jordi Pujols[1], Nunilo Cremades [2✉], Irantzu Pallarès [1✉] & Salvador Ventura [1✉]

α-Synuclein aggregation is a key driver of neurodegeneration in Parkinson's disease and related syndromes. Accordingly, obtaining a molecule that targets α-synuclein toxic assemblies with high affinity is a long-pursued objective. Here, we exploit the biophysical properties of toxic oligomers and amyloid fibrils to identify a family of α-helical peptides that bind to these α-synuclein species with low nanomolar affinity, without interfering with the monomeric functional protein. This activity is translated into a high anti-aggregation potency and the ability to abrogate oligomer-induced cell damage. Using a structure-guided search we identify a human peptide expressed in the brain and the gastrointestinal tract with analogous binding, anti-aggregation, and detoxifying properties. The chemical entities we describe here may represent a therapeutic avenue for the synucleinopathies and are promising tools to assist diagnosis by discriminating between native and toxic α-synuclein species.

[1] Institut de Biotecnologia i Biomedicina and Departament de Bioquímica i Biologia Molecular, Universitat Autònoma de Barcelona, Bellaterra, Barcelona, Spain. [2] Institute for Biocomputation and Physics of Complex Systems (BIFI)-Joint Unit BIFI-IQFR (CSIC), University of Zaragoza, Zaragoza, Spain. ✉email: ncc@unizar.es; irantzu.pallares@uab.cat; salvador.ventura@uab.es

α-Synuclein (αS) is a 140 amino acid protein whose aggregation into amyloid fibrils in a subset of neuronal and glial cells lies behind the onset of a group of progressive and, ultimately, fatal neurodegenerative disorders, including Parkinson's disease (PD)[1–4], that are collectively referred to as synucleinopathies. A causative link between αS and disease is supported by the discoveries that multiplications and missense mutations in SNCA, the αS gene, cause dominantly-inherited familial forms of PD[5].

Interfering with αS amyloid formation and abrogating the associated toxicity is considered a promising therapeutic strategy for synucleinopathies[6–8]. However, the design of molecular entities that target specific αS toxic assemblies is challenging because of the heterogeneous, dynamic, and transient nature of these species. High-throughput screening initiatives have rendered promising αS aggregation inhibitors[9–11]. However, these selection procedures are blind to the ligand mechanism of action. In the absence of a structure-activity relationship, it is difficult to evolve the affinity and specificity of the identified hits to generate drugs that can reach the clinics. The lack of specific and sensitive molecules to detect the pathogenic forms of αS also constrains the early diagnosis of these diseases.

The in vitro aggregation of αS displays a sigmoidal growth profile, suggesting that it follows a nucleation-polymerization mechanism[12], where soluble αS undergoes a nucleation process that produces oligomers able to grow through further monomer addition to form insoluble amyloid fibrils. Oligomeric forms of αS have been detected in the brains and other tissues of patients suffering from PD, and growing evidence suggests that they constitute the primary cytotoxic agents accounting for the gain-of-toxicity associated with αS aggregation, whereas both oligomers and fibrils would be responsible for pathology dissemination in the brain[2,13–15]. We have recently identified the sequential occurrence of two conformationally distinct types of oligomers during αS in vitro fibrillation. The initial non-toxic disordered oligomers, named as type A oligomers, undergo a structural reorganization to form more stable and compact β sheet-enriched, and proteinase K-resistant species that exhibit intrinsic cytotoxicity, named as type B oligomers[16]. Stable, trapped analogues of these two well-defined types of transient oligomers (referred to as type A* and type B* oligomers, where the star refers to the kinetically trapped nature of these isolated oligomeric forms) have been isolated and characterized in detail[13,16] and, therefore, constitute important tools for the development of specific therapeutic and diagnostic strategies.

In this work, we exploit our recent advances in the understanding of the structural determinants of toxicity of αS oligomers to rationally identify peptide molecules able to target αS toxic species. By using a time-resolved single-particle fluorescence approach, we demonstrate that short, amphipathic, and cationic α-helical peptides do not interact with the functional monomeric αS, but they bind toxic oligomers and fibrils with nanomolar affinity, resulting in the substoichiometric inhibition of αS aggregation and abrogation of oligomer-induced damage in neuronal cell models. We then use a protein engineering approach to dissect the molecular determinants accounting for this interaction, which allow us to identify a human peptide, constitutively expressed in the brain and gastrointestinal tract, that binds with low nanomolar affinity to αS toxic assemblies, thus suppressing the aggregation cascade and its associated neurotoxicity. Thus, we describe here the rational identification and characterization of a family of highly potent peptidic ligands able to bind to αS toxic species and abrogate their detrimental effects in neuronal cells. This discovery may open previously unexplored avenues for the diagnosis and/or therapeutics of PD and related disorders.

## Results

**Identification of an αS species-specific peptide ligand.** We rationalized that the particular properties of the four main αS conformers identified during αS amyloid aggregation, namely monomers, non-toxic (type A/A*) oligomers, toxic (type B/B*) oligomers and fibrils, could be exploited to identify a selective ligand for the main species responsible for induction and propagation of toxicity, which are currently believed to be type B-like oligomers and amyloid fibrils, respectively[17]. In Fig. 1a, we illustrate the dissection of the differential traits of αS species (for a more detailed morphological, size and structural characterization of the different αS species in isolated preparations see Supplementary Fig. 1). Type B-like oligomers and amyloid fibrils share two features: (i) they expose relatively large lipophilic clusters to the solvent (Supplementary Fig. 1e). These hydrophobic surfaces induce cellular toxicity and drive subsequent fibrillation[16,18–20]. (ii) They possess a high anionic character at neutral pH, as a result of the stacking of αS monomers (net charge −9). In αS, the negative charge is concentrated at the C-terminal region (residues 95–140), which clusters 15 E/D amino acids. This αS segment remains disordered, and solvent exposed in both oligomers and fibrils[13,21–23], being thus accessible to putative ligands.

While the solvent exposure of hydrophobic surfaces seems to be a general feature of toxic pre-fibrillar oligomers[13,24], the combination of highly exposed hydrophobicity and negative charge is likely unique to these two toxic αS assemblies. Thus, we hypothesized that hydrophobic patches embedded in an anionic environment might delineate a diffuse, but physicochemically-defined, binding surface in these two types of αS aggregates for a complementary molecule; ideally an amphipathic and cationic entity. A short α-helical peptide might provide a structurally stable scaffold to merge both features.

We identified a naturally occurring peptide bearing a short, stable, amphipathic, and cationic helical fold. PSMα3 is a 22-residue bacterial extracellular peptide that has been shown to remain in an α-helical conformation for weeks[25]. It has a net charge of +2, a mean hydrophobicity ($H$) of 0.54, an α-helical hydrophobic moment ($\mu_H$) of 0.56, and the helical wheel plot evidences its amphipathic character (Fig. 1b, c). The far-UV circular dichroism (CD) spectrum of PSMα3 confirms that it folds into an α-helix under our assay conditions (Fig. 1d). Thus, according to our hypothesis, this peptide fulfills all the requirements to bind specifically to type B-like αS oligomers and amyloid fibrils.

We engineered PSMα3 to obtain a negative control peptide in which the formation of an α-helix is strongly disfavored. This will disrupt the peptide amphipathic character and, theoretically, abolish binding to αS type B* oligomers and amyloid fibrils. After a computational proline scanning of PSMα3 using the AGADIR algorithm[26] (Supplementary Fig. 2a), we selected the K9P and F11P mutations, as they have a significant impact in helical propensity and map to opposite faces of the α-helix (Supplementary Fig. 2b, c). The characterization of the secondary structure of the K9P-F11P PSMα3 peptide (further referred to as disrupted PSMα3 or dPSMα3) in solution by CD confirmed the disruption of the α-helix fold (Supplementary Fig. 2d). Thus, dPSMα3 constitutes a suitable negative control for further studies, as it keeps a sequence identity of 91% with PSMα3, but lacks its amphipathic character, a feature that we propose is key for the species-specific binding to the αS toxic assemblies.

**Selective interaction of PSMα3 with αS toxic species.** We then addressed the interaction of PSMα3 and dPSMα3 with the above described four αS species. As multiple peptides are expected to bind multiple αS molecules in the aggregated states, the binding

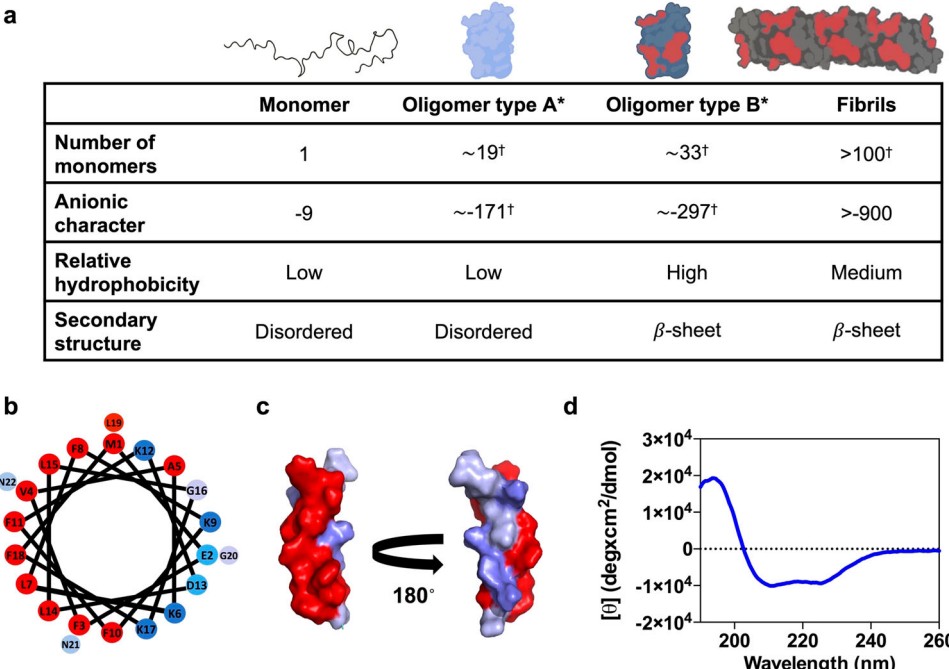

**Fig. 1 Rational identification of a peptide ligand for αS toxic species. a** Main molecular features of the four isolated α-synuclein (αS) species. Values with a dagger (†) represent extrapolations based on the average number of monomers in each species. In the upper schemes of αS oligomeric and fibrillar species, the acidic C-terminal region is not depicted since it has been described to be in a disordered and conformationally flexible state[21–23]. **b** Helical wheel projection of PSMα3 sequence (red, hydrophobic residues; blue pallet, hydrophilic residues depending on their character). **c** Surface representation of the three-dimensional structure of PSMα3 with hydrophobic residues in red and hydrophilic residues in blue. **d** Far-UV circular dichroism spectra of PSMα3.

process can only be well described if both the stoichiometry of the complex and the affinity of the peptide for the αS molecules in a particular conformer is known. In order to obtain good estimates of both parameters, we exploited the power of dual-color fluorescence cross-correlation spectroscopy (dcFCCS), a time-resolved fluorescence fluctuation technique that allows the direct observation of co-diffusing fluorescent species arising from interactions between differently labeled molecules or assemblies in solution[27,28]. To this end, αS species were cystein-labeled with maleimide-AlexaFluor488 (AF488), with each αS molecule of the different species containing one fluorophore at position 122, and the peptides were cystein-labeled with maleimide-Atto647N at the N-terminus (see "Methods" for details). Simultaneously, we assessed complex formation by single-particle fluorescence spectroscopy (SPF) analysis, including Förster Resonance Energy Transfer (FRET) and donor/acceptor stoichiometry (S), in order to validate and complement the dcFCCS approach. These approaches allow us to monitor distinct individual species simultaneously by avoiding measurements of ensemble averages (Supplementary Fig. 3) and have been previously used for the characterization of αS aggregation pathways and the study of αS interactors[29–31].

We first assessed the binding of PSMα3 to monomeric αS and found by dcFCCS analysis that these molecules were unable to interact when mixed in an equimolar ratio (even at concentrations as high as 15 nM of each molecule) (Fig. 2a), as reflected by a flat cross-correlation curve comparable to that of the negative control of cross-correlation (Supplementary Fig. 4). We then analyzed the interaction of PSMα3 with the different αS aggregated species by means of dcFCCS at approximately equimolar ratios of peptide and αS molecules. Due to the different stabilities of the various aggregated species upon single-molecule dilution and their differential adsorption to the surface

of the coverslips, the total αS concentration of each aggregate sample was adjusted between 1 and 5 nM (in mass concentration) so that the frequency of events in the measurements was very similar between the various aggregated samples. It is important to note that, for αS aggregated species, consisting of several tens of monomers, the species concentrations are in the picomolar range and, as further explained in the "Methods", single-particle conditions are ensured throughout the experiments. Under these conditions, a marginal cross-correlation amplitude was observed for the type A* oligomers (Fig. 2b), whereas a clear cross-correlation was obtained for the interaction of the peptide with both type B* oligomers and fibrils (Fig. 2c and d, respectively), already indicating a stark difference in the binding ability of the peptide to the different aggregates. Consistently, single-particle burst-wise analysis revealed a high number of FRET events, thus validating the direct interaction of the peptide with these two αS species (Supplementary Fig. 5a, b). In contrast, the same analysis yielded either a statistically insignificant number of FRET events or none at all for the interaction between PSMα3 and αS type A* oligomers and monomers, respectively (Supplementary Fig. 5c). These results offer a single-particle understanding of a complex binding scenario and further reinforce the observations derived from dcFCCS experiments.

We next performed a titration experiment for analyzing the binding of the peptide to the three αS aggregated species. For this, we developed a model-independent saturation binding curve, based on the theoretical framework previously developed by Kruger and coworkers[32]. This analysis allowed us to quantify the number of peptide molecules bound to each αS species ($N_P$) as a function of the concentration of unbound peptide ($C_{p,free}$) (Fig. 2e). Using a simplistic Langmuir isotherm model, we estimated the single-state dissociation constant ($K_D$) of the interactions and the average maximum number of peptide

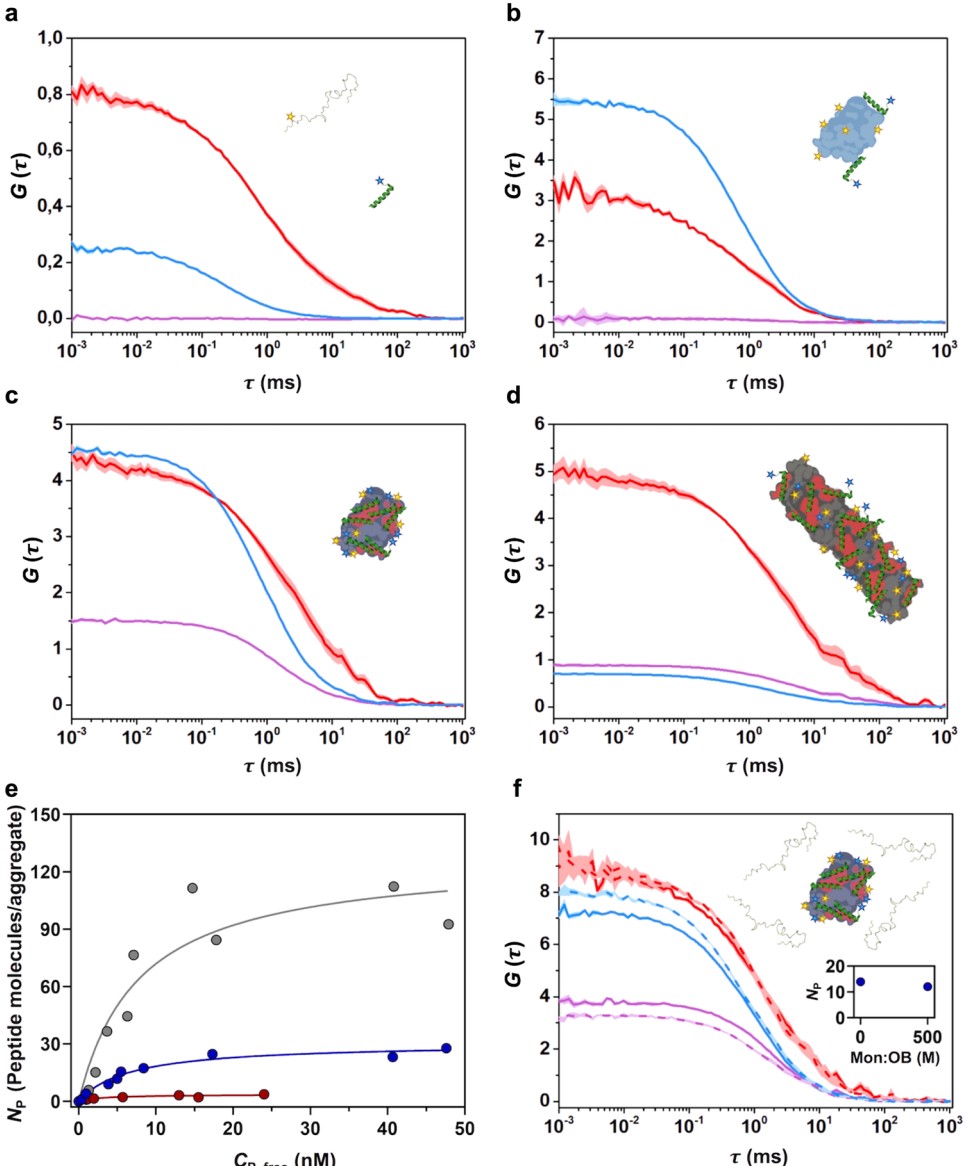

**Fig. 2 Interaction of PSMα3 with different αS species by dcFCCS. a–d** Representative auto-correlation curves for α-synuclein (αS) (blue line) and PSMα3 (red line) and cross-correlation curves for interacting molecules (purple line). The amplitude (G) error is shown as faint colored area for the corresponding correlation curves. Samples contained (**a**) ~15 nM αS monomer and ~15 nM PSMα3, (**b**) 1 nM type A* and ~5 nM PSMα3, (**c**) 1 nM type B* oligomers and ~5 nM PSMα3 or (**d**) ~5 nM sonicated fibrils and ~5 nM PSMα3. **e** Titration binding curves for the interaction of PSMα3 with type A* oligomers (red circles), type B* oligomers (blue circles) or sonicated fibrils (gray circles) obtained by dcFCCS, showing their corresponding analysis assuming a model of $n$ identical and independent binding sites (referred in equation 7 as $N_{max}$) per αS aggregated species (solid lines). $N_P$ represents the number of bound peptides per aggregate. **f** Auto-correlation curves (αS in blue, PSMα3 peptide in red) and cross-correlation curve for the interacting molecules (in purple) obtained in samples containing ~1 nM αS type B* oligomers and ~2 nM PSMα3 in the absence (solid lines) or presence (dashed lines) of a 500-molar excess of unlabeled monomer with respect to the particle concentration of oligomers. The inset shows the number of bound peptides ($N_P$) per aggregate in both conditions. For αS aggregated species, each consisting of several tens of monomers, the species concentrations are in the picomolar range and, as further explained above, single-particle conditions are ensured throughout the experiments.

binding sites ($N_{max}$) in each type of αS species. Interestingly, while the $K_D$ values for the peptide-αS interaction obtained for type A* oligomers, type B* oligomers and fibrils are very similar, in all cases in the very low nM range (3.07 nM, 6.67 nM, and 7.8 nM, respectively), the average maximum number of peptides per aggregate ($N_P$) varies remarkably, being 3, 30, and 120, respectively. This indicates that the main difference between the three aggregated species in terms of PSMα3 interaction is the number of binding sites per aggregate rather than the affinity of the peptide for them. Of note, the average maximum number of binding sites obtained for the type B* oligomers and fibrils nearly

matches the average number of αS molecules per aggregate species (19, 33, and 107 for type A*, type B* oligomers, and fibrils, respectively, estimated by comparing the molecular brightness of the aggregated species to that of the αS monomer), while for the type A* oligomers represents only one sixth of the average αS molecules in this type of aggregate. This data is in agreement with the single-particle fluorescence analysis obtained for the different complexes, where decreasing fluorescence stoichiometry values are found for the interacting pairs with increasing PSMα3 concentrations (Supplementary Fig. 6) yielding a binding curve remarkably similar to that obtained by dcFCCS

(Supplementary Fig. 7). In addition, only very few FRET events were observed for the binding of PSMα3 to type A* oligomers, in contrast to the numerous FRET events with a defined FRET efficiency (E) distribution observed for the binding to type B* oligomers and fibrils (Supplementary Fig. 5). Together, the dcFCCS and single-particle fluorescence spectroscopy data demonstrate that PSMα3 is a high affinity ligand of αS toxic species, with affinities in the low nanomolar range, and with a high avidity for the toxic αS species, namely the type B-like oligomers and fibrils. Note that a 500-fold molar excess (in monomer equivalents) of unlabeled monomeric αS does not interfere with the binding of PSMα3 to type B* oligomers (Fig. 2f), which indicates a high specificity towards toxic aggregated species and negligible monomer binding, a feature difficult to find in previously reported αS ligands.

It might be important to indicate that PSMα3 could present a certain degree of oligomerization that results in a slower diffusion than expected for a peptide monomer ($14.6 \pm 3.6\ \mu m^2\ s^{-1}$, Fig. 2). Despite this, our dcFCCS-derived binding curves indicate that the monomeric form of the peptide can effectively bind the αS aggregated species with $N_P$ values as low as 1 (further information is provided in the "Methods").

Interestingly, when we analyzed the binding of the PSMα3 analogous, but disordered peptide, dPSMα3, we could not detect any interaction with any of the four αS species (Supplementary Fig. 8), indicating that an amphipathic distribution of the peptide residues, achieved through an α-helical conformation, is a requirement for the interaction.

Overall, our dcFCCS and single-particle fluorescence spectroscopy-derived binding analysis indicates that PSMα3 binds with low nanomolar affinity to αS aggregated species. The degree of binding is limited by the number of available interaction sites, which is likely associated with the extent of solvent-exposed hydrophobic surface per aggregate, which depends on both the size and the lipophilicity of the aggregate, in agreement with our initial reasoning.

**PSMα3 inhibits αS amyloid aggregation.** We hypothesized that the high affinity and number of binding sites of type B* oligomers for PSMα3 might result in the partial or full coverage of the surface of these assemblies, as well as their structurally homologous type B oligomers, thus preventing their progression to fibrils during the αS amyloid aggregation process. To assess if this was the case, we set up in vitro αS aggregation reactions in the absence and presence of an equimolar concentration of PSMα3 (70 μM) and followed its progression by monitoring the increase in thioflavin-T (Th-T) fluorescence. After 32 h of incubation, a ~90% decrease in Th-T fluorescence emission, relative to the untreated sample, was observed in the presence of PSMα3, suggesting that the peptide acts as a potent inhibitor of αS amyloid aggregation (Fig. 3a). Inhibition was orthogonally confirmed by quantifying the fraction of αS that remains soluble at the endpoint of the reaction spectroscopically and by SDS-PAGE (Supplementary Fig. 9a, b). The inhibitory activity of PSMα3 was concentration-dependent and significant inhibition was observed even at a substoichiometric 20:1 ratio (αS:PSMα3) (Fig. 3b). Transmission electron microscopy (TEM) images confirmed that samples incubated with PSMα3 contained very few fibrils per field, in comparison to untreated samples (Fig. 3c). The observation that dPSMα3 exhibited a negligible anti-aggregative activity (Supplementary Fig. 10) reinforces the connection between the binding of the amphipathic (hydrophobic/cationic) helical peptide to αS oligomers and its potent amyloid inhibition activity.

PSMα3 is a better inhibitor of αS amyloid aggregation than SynuClean-D (Fig. 3a), a small molecule with high

neuroprotective activity in *Caenorhabditis elegans* models of PD that we have recently discovered[9].

To gain further information on the inhibitory mechanism and pinpoint the αS species targeted along the complex pathway of aggregation, we isolated the low molecular weight species generated at the early stages of aggregation (see "Methods"). Electron microscopy analysis revealed that in control aggregation reactions, after 12 h of incubation, αS mainly populates small fibrillar species and round prefibrillar aggregates (average diameter between 20 and 40 nm) (Fig. 3d and Supplementary Fig. 11a–b). In contrast, at the same time point, samples incubated with PSMα3 contained a large fraction of small oligomers of annular shape with diameters between 9 and 14 nm, morphologically similar to type B* oligomers (Fig. 3d and Supplementary Fig. 11c) and other annular oligomers previously described in the literature[24,33,34]. Together with the time-resolved fluorescence spectroscopy data, this evidence strongly suggests that PSMα3 could be preventing or retarding the conversion of type B-like oligomers into fibrillar species. Notably, this result endorses the use of the kinetically stabilized type B* oligomers as mimicries of the toxic oligomers that populate αS aggregation reactions.

**PSMα3 protects cells from αS oligomer-induced cell damage.** As it occurs with toxic oligomers from other amyloidogenic systems, the toxicity of type B* oligomers relies on their ability to interact and disrupt cellular membranes[13]. In αS this activity is encoded in two of their characteristic structural elements[13]: (i) an exposed N-terminal region that acts as the initial anchor to the membrane surface, similarly as with the monomeric functional form of the protein, and (ii) a β-sheet core, composed primarily by the central region of the protein, with significant hydrophobic surface exposed to the solvent that then inserts into the lipid bilayer causing major perturbations. The highly negatively charged C-terminal region of the protein remains disordered without significant interactions with the membrane.

We hypothesized that the binding of PSMα3 to type B* oligomers, mediated in part by the solvent-exposed hydrophobic regions of the β-sheet core, would block their exposed lipophilic elements, thus decreasing its ability to insert into and perturb the membrane bilayer and induce cellular toxicity. We treated human SH-SY5Y neuroblastoma cells with 10 μM of oligomers and observed that, as previously reported[35], they possess a high affinity for cellular membranes (Fig. 4a–b). When the oligomers were preincubated with an equimolar concentration of PSMα3, we observed a ~60% reduction in the amount of αS bound to cells relative to untreated oligomers, indicating that PSMα3 binding to type B* oligomers directly affects the binding of the oligomers to cellular membranes. As expected, pretreatment of the oligomers with dPSMα3 did not interfere with their interaction with cells.

One of the earliest effects of type B* oligomer-mediated membrane perturbation is the substantial increase in the levels of intracellular reactive oxygen species (ROS)[24], which in turn elicits mitochondrial dysfunction[36]. We assessed if the blockage of the oligomer regions involved in membrane perturbation by PSMα3 binding could protect membrane integrity and therefore prevent its associated increase in intracellular ROS levels. Treatment of neuroblastoma cells with 10 μM of oligomers induced a drastic increase in ROS levels (Fig. 4c–d). However, when these oligomers were preincubated with equimolar (1:1) and substoichiometric (1:0.2) concentrations of PSMα3, the ROS levels of treated cells approached those of healthy, untreated, cells, indicating that PSMα3 protects against oligomers-induced damage. This detoxifying activity seems to be associated with the particular structural and physicochemical properties of this

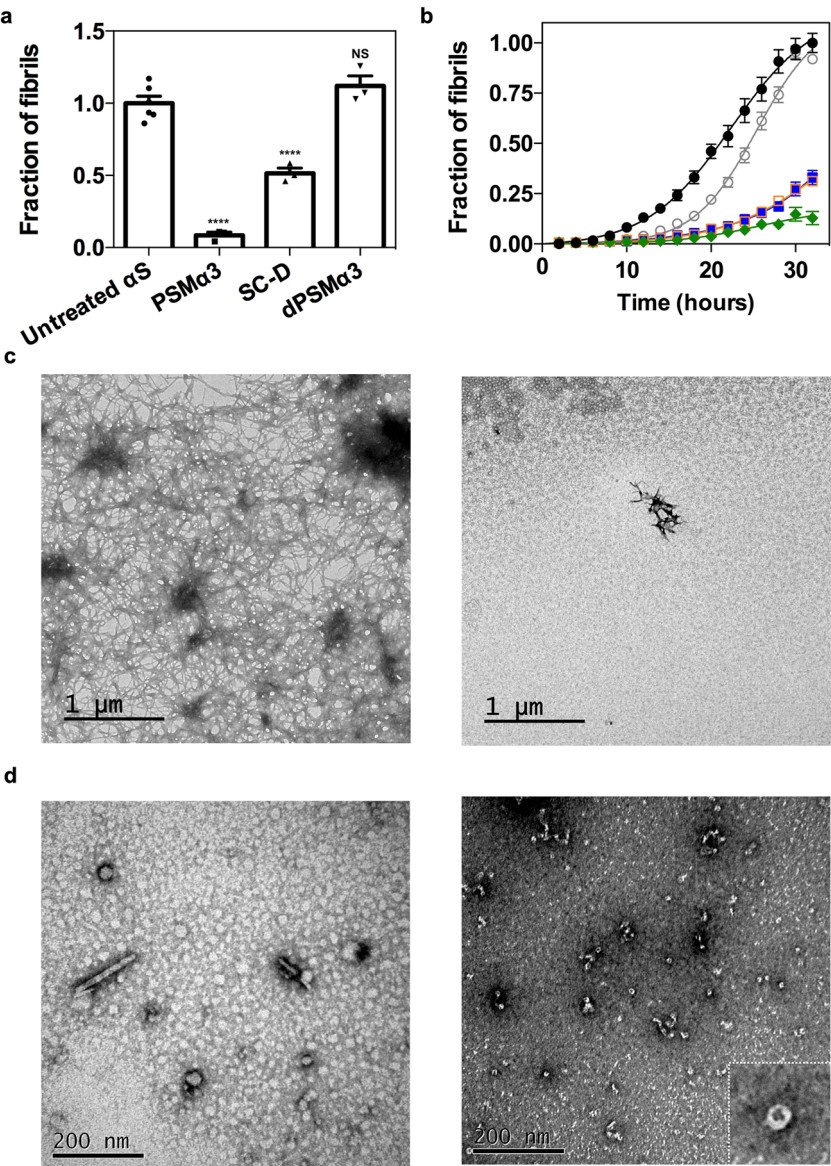

**Fig. 3 Effect of PSMα3 on in vitro αS amyloid fibrillation. a** Inhibition of α-synuclein (αS) amyloid aggregation as measured by Th-T fluorescence after 32 h incubation in the presence of equimolar concentrations of PSMα3, SynuClean-D (SC-D), and dPSMα3. **** $p < 0.0001$ relative to untreated αS (unpaired two-tailed $t$ tests (Welch-corrected)). Data were expressed as mean ± s.e.m ($n = 6$ and 3 independent experiments for untreated and treated conditions respectively). *NS* no significant, $p = 0.23$. **b** Aggregation kinetics of 70 μM αS and titration of the inhibitory activity of PSMα3 at different concentrations: 35 μM (green), 14 μM (orange), 7 μM (blue), 3.5 μM (gray) and in the absence of PSMα3 (black). Data were expressed as mean ± s.e.m ($n = 9$ independent experiments). **c** Representative TEM micrographs of αS aggregated for 32 h in the absence (left) and presence of an equimolar concentration of PSMα3 (right) that came from two independent replicates. **d** Representative TEM micrographs illustrating the morphological differences between low-molecular weight aggregates of αS after 12 h of incubation in absence (left) and presence of PSMα3 (right). Results are consistent in two independent replicates. Inset shows an annular oligomer at high magnification.

peptide since treatment with equimolar concentrations of dPSMα3 failed to exert any protective effect.

**Dissection of PSMα3 aggregation-inhibitory determinants**. To this point, we have assigned the αS binding, anti-aggregation and cytoprotective properties of PSMα3 to its helical, amphipathic and cationic character. To confirm that this is the case, we reverse-engineered PSMα3 into a non-natural peptide scaffold with low sequence complexity that keeps its critical properties. We employed a set of bioinformatics tools to predict the helical propensity, helical hydrophobic moment, and thermodynamic stability of our successive designs using AGADIR[26], HELIQUEST[37], and FOLDX[38], respectively. Data regarding those

predictions are displayed in Supplementary Table 1. Then, we evaluated the anti-aggregative properties of these molecules, under the assumption that the inhibitory capacity is connected with the oligomer-peptide interaction affinity.

A first requirement for binding is a continuous hydrophobic face to interact with the surface of oligomers or fibrils. In our view, the specific sequence of this helical side would be irrelevant, as long as it keeps its lipophilic character. To demonstrate that this assumption is correct, we mutated all the residues in the hydrophobic face of the PSMα3 α-helix to leucine (All_Leu), generating an amphipathic peptide devoid of any sequence diversity in this side. Simultaneously, we designed a variant of All-Leu devoid of the last three C-terminal residues (All_Leu19) since they are not part of the α-helix,

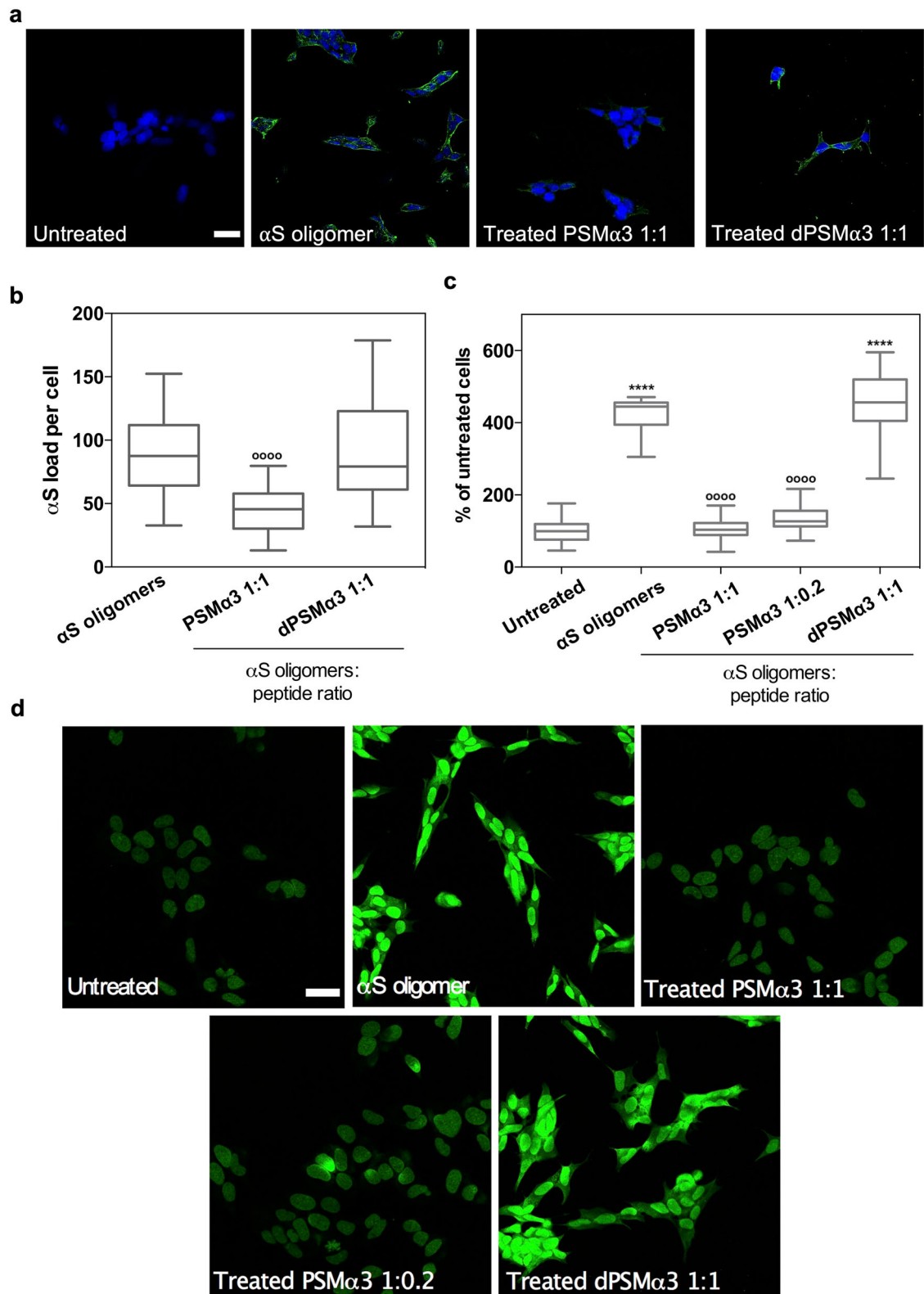

and thus they are not expected to contribute significantly to the binding. Both redesigned peptides folded into α-helices and retained the inhibitory activity, with a potency that approaches that of PSMα3 (Fig. 5a–b). Thus, we concluded that it is the generic hydrophobic character of the helical face and not its sequence or composition that is relevant for the binding.

Next, to further reduce the peptide sequence complexity, we redesigned the hydrophilic face in such a way that it only contained

ionizable residues. We designed a new variant (Scaffold_19) with only four different amino acids (Leu, Asp, Glu, and Lys) by introducing two point-mutations (A5E_G16K) in the All_Leu19 peptide. We decided to maintain the peptide net charge by introducing residues with opposed charges. Thus, Scaffold_19 only has Leu in the hydrophobic face and charged residues in the hydrophilic one. This variant folded into an α-helix and showed the same anti-aggregation activity than the parental variant (Fig. 5c).

**Fig. 4 Suppression of the αS oligomers-induced damage in neuroblastoma cells. a** Representative confocal images showing the α-synuclein (αS) load per cell after the treatment with 10 μM of type B* oligomers pretreated with an equimolar concentration of PSMα3 or dPSMα3. Scale bar represents 30 μm. **b** Quantification of the αS load per cell. ****$p < 0.0001$ relative to untreated cells. °°°°$p < 0.0001$ relative to cells treated with αS type B* oligomers. Unpaired two-tailed $t$ tests (Welch-corrected). 76, 72, and 67 cells, (respectively, for αS oligomers, PSMα3 1:1, PSMα3 1:02 and dPSMα3) were analyzed from two independent experiments. **c** Quantification of the levels of intracellular ROS of SH-SY5Y cells incubated with 10 μM of type B* oligomers preincubated with different concentrations of PSMα3 or dPSMα3. °°°°$p < 0.0001$ relative to cells treated with αS type B* oligomers. (Unpaired two-tailed $t$ tests (Welch-corrected)). 233, 230, 240, 212, and 100 cells, (respectively, for untreated, αS oligomers, PSMα3 1:1, PSMα3 1:02 and dPSMα3) were analyzed from two independent experiments **d** Representative confocal images of the analysis of panel (**c**). Scale bar represents 30 μM. In (**b**) and (**c**) data are represented as box and whiskers plots where the middle line is the median, the lower and upper hinges correspond to the first and third quartiles, the upper whisker extends from the hinge to the largest value no further than 1.5 × IQR from the hinge (where IQR is the inter-quartile range) and the lower whisker extends from the hinge to the smallest value at most 1.5 × IQR of the hinge.

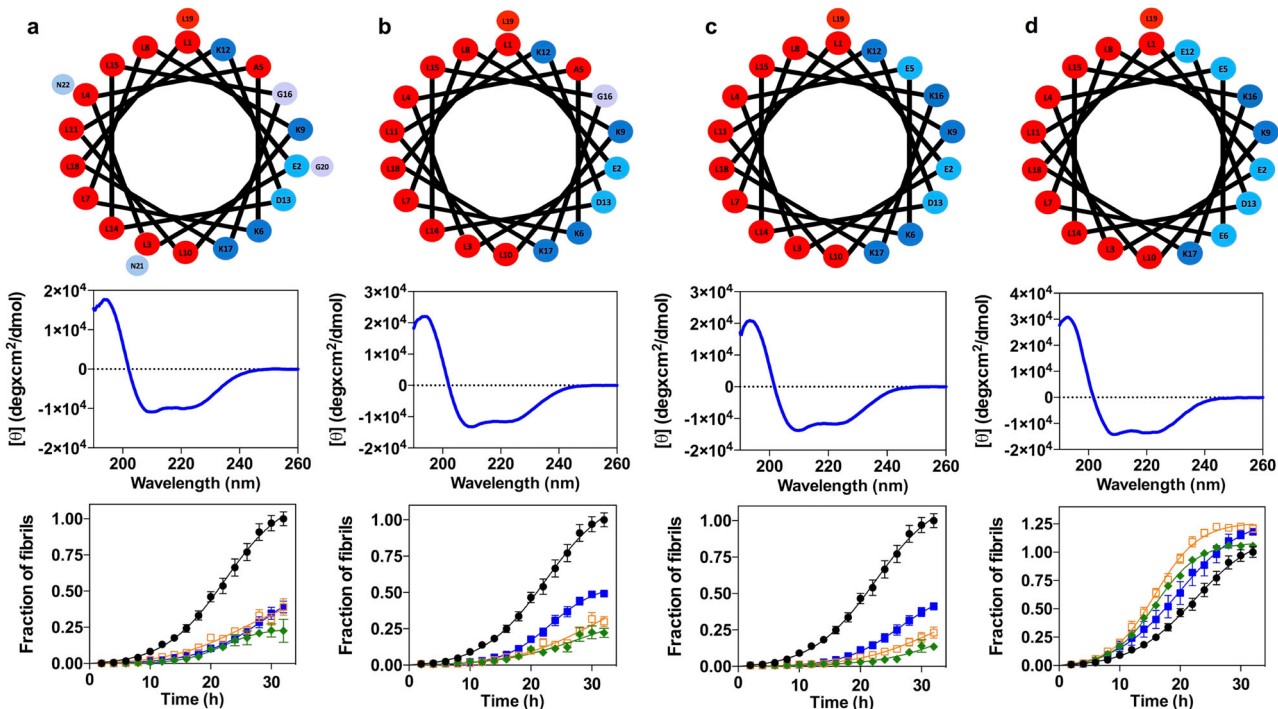

**Fig. 5 Redesign of PSMα3 variants to dissect the molecular determinants of the anti-aggregative activity. a–d** Helical wheel (red, hydrophobic residues; blue pallet, hydrophilic residues) (up), circular dichroism spectra (mid) and titration of the inhibitory activity of PSMα3 variants at different concentrations (down): 35 μM (green), 14 μM (orange), 7 μM (blue) and in the absence of PSMα3 variants (black). Variants: All_Leu (**a**), All_Leu19 (**b**), Scaffold_19 (**c**), and Anionic_scaffold (**d**). Data were expressed as mean ± s.e.m ($n = 9$ independent experiments).

The simplicity of Scaffold_19 allowed us to redesign the net charge of the peptide, to validate the other physicochemical property theoretically contributing to binding: a net positive charge. We generated a peptide with a net charge of −2 by introducing two charge-reversing mutations (K6E_K12E). This anionic peptide (Anionic_scaffold) folds into an α-helix and has a helical hydrophobic moment ($\mu_H$) of 0.65, indicative of an amphipathic nature, but does not inhibit αS amyloid aggregation, confirming that a cationic character in the hydrophilic face is a requirement for binding (Fig. 5d).

Overall, we succeeded in dissecting the peptide features responsible for aggregation inhibition. In the process, we generated, Scaffold_19, a short peptide with low sequence complexity whose inhibitory activity does not stem from the primary sequence, but instead from a defined spatial distribution of two physicochemical traits.

**LL-37 inhibits αS aggregation and oligomers-induced cell damage**. Once we elucidated the determinants of this mechanism of αS amyloid inhibition, we wondered if this activity could also

be encoded in natural human peptides. First, we screened the EROP-Moscow oligopeptide database[39] for human cationic peptides longer than 10 residues (≥3 helical turns), obtaining 287 hits. Next, we run AGADIR on them to exclude peptides with a low helical propensity, which resulted in 25 peptides, from which only 9 peptides were predicted to have at least a partial amphipathic character, according to their helical hydrophobic moment ($\mu_H$) (Supplementary Table 2). Then we screened the literature for candidates whose tissue distribution overlapped with that of αS and selected LL-37, the only human member of the cathelicidin family of antimicrobial peptides, for its further characterization. LL-37 is a 37-residue peptide resulting from a post-translational cleavage at the C-terminus of cathelicidin hCAP18[40]. This peptide is constitutively expressed in the brain and the gastrointestinal tract; its presence in both tissues is engaging, as the brain-gut axis connection is gaining momentum in PD[41–43].

First, we confirmed that LL-37 adopts an α-helical conformation under our assay conditions (Fig. 6a, b). With α-helical hydrophobic moment ($\mu_H$) of 0.52, the helical-wheel projection and the available 3D-structures[37] indicate that this α-helix would be both cationic and amphipathic (Fig. 6a, b). Then, we titrated

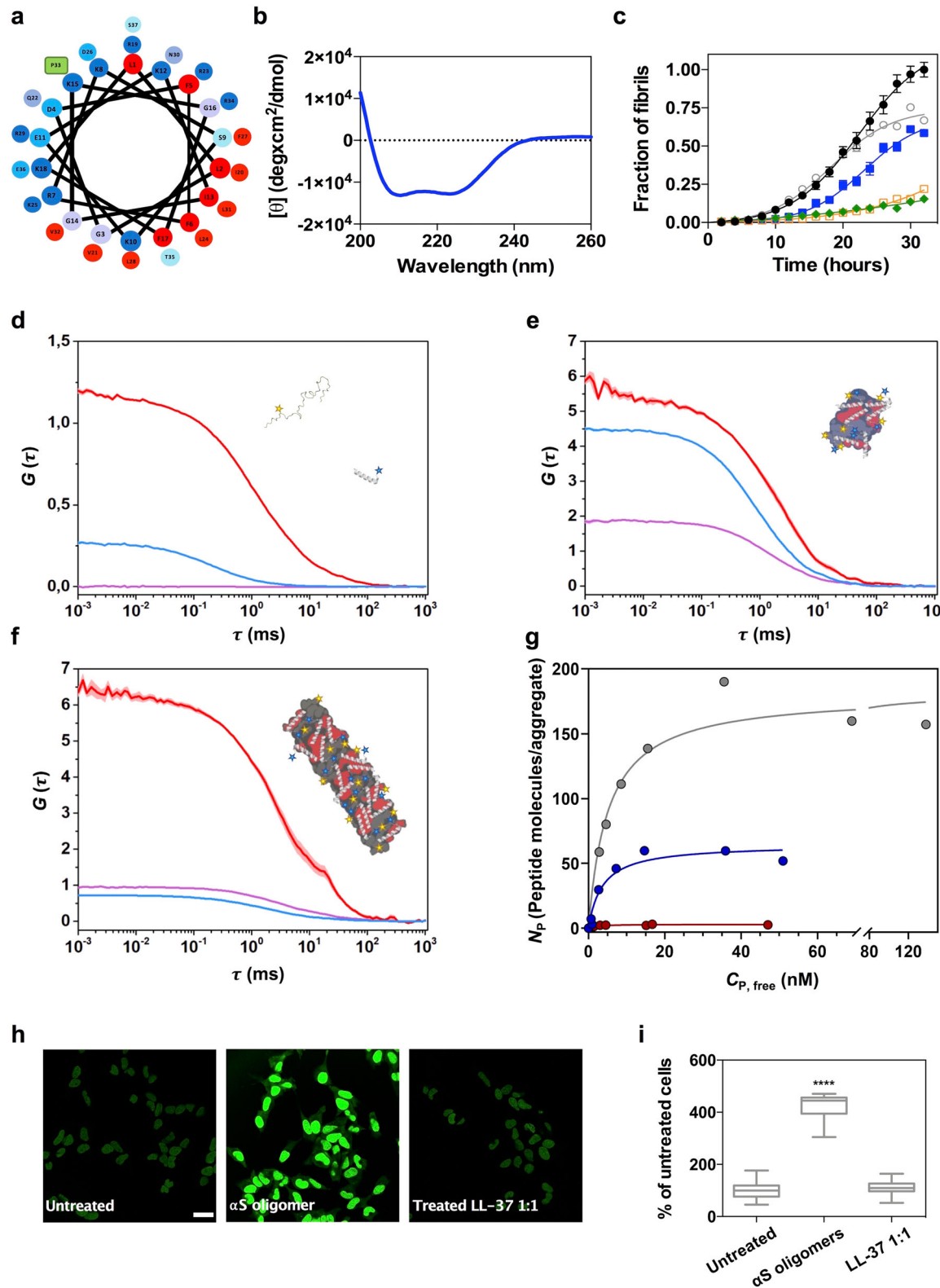

the anti-aggregative activity of LL-37, confirming that it suppresses αS amyloid formation at substoichiometric concentrations (Fig. 6c). Sedimentation analysis provides orthogonal support for the αS aggregation inhibitory activity of LL-37 (Supplementary Fig. 9). Next, we labeled LL-37 with maleimide-Atto647N at a single engineered cysteine at the N-terminus, and we performed time-resolved dual-color fluorescence spectroscopy experiments as described previously for PSMα3. Both dcFCCS

and spFRET (Fig. 6d, f and Supplementary Fig. 12) reported a strong binding to type B* oligomers and fibrils, a weak interaction with type A* oligomers and the absence of any interaction with the αS monomer, indicating that LL-37 and PSMα3 share a very similar binding mechanism (Fig. 6g). In this case, LL-37 displays slightly higher affinities than PSMα3 ($K_D = $ 3.62 nM for type B* oligomers, $K_D = $ 5.14 nM for sonicated fibrils and $K_D = $ 1.92 nM for type A* oligomers), and a significantly

**Fig. 6 Characterization of the interaction of LL-37 with the αS toxic species. a** Helical wheel projection of LL-37 sequence (red, hydrophobic residues; blue pallet, hydrophilic residues; green, proline). **b** Far-UV circular dichroism spectra of LL-37 in PBS pH 7.4. **c** Aggregation kinetics of 70 μM α-synuclein (αS) and titration of the inhibitory activity of LL-37 at different concentrations: 35 μM (green), 14 μM (orange), 7 μM (blue), 3.5 μM (gray) and in the absence of peptide (black). Data were expressed as mean ± s.e.m ($n = 9$ independent experiments). Representative auto-correlation curves for αS and LL-37 peptide and cross-correlation curves for interacting molecules are shown in blue, red and purple lines, respectively. The amplitude ($G$) error is shown as faint colored area for the corresponding correlation curves. Samples contained (**d**) ~15 nM αS monomers and ~15 nM LL-37, (**e**) 1 nM type B* oligomers and ~5 nM LL-37 or (**f**) ~5 nM sonicated fibrils and ~5 nM PSMα3. **g** Titration binding curves for the interaction of LL-37 with type A* oligomers (red circles), type B* oligomers (blue circles) or sonicated fibrils (gray circles) obtained by dcFCCS, showing their corresponding analysis assuming a model of $n$ independent binding sites per αS aggregated species (solid lines). $N_P$ represents the number of bound peptides per aggregate. **h** Representative confocal images of SH-SY5Y cells treated with 10 μM of type B* oligomers in the presence of an equimolar concentration of LL-37. Scale bar represents 30 μm. **i** Quantification of the intracellular ROS of the experiment displayed in panel (**h**). ****$p < 0.0001$ relative to untreated cells. °°°°$p < 0.0001$ relative to cells treated with αS type B* oligomers. Unpaired two-tailed $t$ tests (Welch-corrected). A total of 233, 230, and 199 cells, (respectively, for untreated αS oligomer and LL-37 1:1) were analyzed from two independent experiments. For αS aggregated species, consisting of several tens of monomers, the species concentrations in (**d**–**g**) are in the picomolar range and, as further explained above, single-particle conditions are ensured throughout the experiments. In (**i**) data are represented as box and whiskers plots where the middle line is the median, the lower and upper hinges correspond to the first and third quartiles, the upper whisker extends from the hinge to the largest value no further than 1.5 × IQR from the hinge (where IQR is the inter-quartile range) and the lower whisker extends from the hinge to the smallest value at most 1.5 × IQR of the hinge.

higher number of binding sites in type B* oligomers ($N_{max} = 64$) and sonicated fibrils ($N_{max} = 181$), while remain the same as PSMα3 for the number of binding sites in the type A* oligomers, which was in any case marginal ($N_{max} = 3$). Consistent with the LL-37 ability to bind type B* oligomers with high affinity, the preincubation of these toxic species with the human peptide at an equimolar concentration completely abolished the production of ROS in neuroblastoma cells (Fig. 6h, i). LL-37 is not related in sequence to PSMα3 or Scaffold_19, but the three peptides share the same structural and physicochemical traits. This confirms that a linear combination of these properties suffices to identify, and potentially design, potent inhibitors of αS aggregation.

Whether LL-37 is actually involved or not in the pathogenesis of PD remains unexplored. However, it is tempting to speculate that small peptides able to interact actively with αS aggregated species might cohabitate with this protein in tissues relevant to the disease. These human peptides may open an unexplored avenue for PD treatment, i.e., by stimulating their endogenous expression.

## Discussion

Because of its involvement in PD and other synucleinopathies, αS aggregation remains a promising target for therapeutic intervention. Herein, we propose a strategy for targeting the αS species behind the onset of these neurodegenerative diseases selectively. By binding to αS toxic oligomers and fibrils, the described collection of peptides inhibits the progression of αS aggregation, while suppressing oligomer mediated cell damage. Importantly, because the binding determinants are structurally encoded, these peptidic molecules do not recognize monomeric αS. Furthermore, the avidity of these peptides for early non-toxic oligomers is more than one order of magnitude lower than the one for type-B* oligomers and fibrils, indicating that they are very selective for these toxic species.

We describe here, non antibody-related biomolecules targeting αS aggregated species, which have been rationally predicted, identified, and engineered. PSMα3 is a first-in-class hit molecule that sets the ground for the future advancement of a generation of leads for disease modification in PD and other synucleinopathies. The requisites for a high peptide binding affinity and αS toxic species selectivity are relatively simple: hydrophobic and positively charged surfaces with opposed orientations in space. This is best exemplified by Scaffold_19, a short and low complexity peptide that fulfills those conditions. This defined binding mode should help in the development and diversification of ligands with increased activities.

Many small bioactive peptides are derived from larger precursors and generated after proteolytic cleavage[44]. In some cases, these peptides are encrypted inside globular proteins, and their processing results in the manifestation of a new biological function. LL-37 is a cathelicidin-derived peptide constitutively expressed in the human brain[45]. Here, we show that LL-37 is a tight binder of αS toxic assemblies, with anti-aggregation and cytoprotective properties. LL-37 has been reported to inhibit the amyloid aggregation of two other disease-linked peptides, Aβ-42[46] and IAPP[47]. However, the mechanism behind this activity is different from the one we describe here since it relies on a certain degree of sequence homology between short linear stretches in these molecules, with LL-37 binding to both the monomers and the aggregated species[46,47]. The $K_{Ds}$ for the binding of LL-37 to freshly resuspended and 24-days-incubated Aβ-42 peptide are 13.30 μM and 20.30 μM, respectively[46]; thus, several orders of magnitude weaker than the ones we report here for αS. Irrespective of the affinity and species selectivity, it is tempting to suggest that endogenous peptides, similar to those described here, could compose a regulatory system where they act as silent guardians of the proteome by targeting aggregation-prone proteins.

Apart from potential future therapeutic implications, the ability of the amphipathic cationic helical peptides to bind to αS toxic species with high affinity might find a direct application in diagnosis. The presence of αS aggregates in biofluids is considered a biomarker for PD and other synucleopathies[48,49]. However, current detection methods are not specific and sensitive enough for their clinical implementation. For instance, ELISA approaches based on the so-called conformation-specific antibodies, perform better when the detection is normalized relative to the total levels of αS or when the same epitope-blocking antibody is used for both capture and detection[50]. This indicates that a major limitation of these methods is the unwanted cross-reaction of the antibodies with the large excess of αS monomer in the fluid. The peptides we describe here do not interact with monomeric αS, and the presence of up to 500-fold molar excess of monomeric αS does not interfere with the detection of nanomolar amounts of toxic oligomers. This property, together with their close-to-antibody affinities, may turn useful for diagnostic purposes. We envision that a strategy which combines sequence-specific αS ligands (i.e., available antibodies) with our species-specific peptides might succeed in the selective and sensitive detection of toxic αS species in biological fluids.

Overall, the molecular entities we describe in this work may help to develop therapeutic and diagnostic strategies for the synucleinopathies.

## Methods

**αS expression and purification.** *Escherichia coli* BL21 (DE3) cells containing a pET21a plasmid encoding the αS gene were grown in LB medium supplemented with 100 μM/mL ampicillin. Protein expression was induced at an optical density of 0.8 (600 nm) with 1 mM isopropyl β-D-thiogalactopyranoside (IPTG) for 4 h. Cells were harvested by centrifugation and washed up by resuspension and centrifugation in PBS pH 7.4. Next, pellets were resuspended in 50 mL per culture liter in lysis buffer (50 mM Tris pH 8, 150 mM NaCl, 1 μg/mL pepstatin, 20 μg/mL aprotinin, 1 mM benzamidine, 1 mM PMSF, 1 mM EDTA, and 0.25 mg/mL lysozyme) and sonicated using a LabSonic®U sonicator (B. Braun Biotech International, Melsungen, Germany). Samples were boiled during 10 min at 95 °C and centrifugated at 20,000 g at 4 °C for 40 min. The soluble fraction was treated with 136 μL/mL of 10% w/v streptomycin sulfate and 228 μL/mL of pure acetic acid. Upon centrifugation, soluble extracts were fractionated by adding 1:1 of saturated ammonium sulfate and resuspending the insoluble fraction with 50% ammonium sulfate. The pellet was resuspended in 100 mM pH 8 ammonium acetate (5 mL per culture liter) and pure EtOH 1:1 (v/v) and harvested by centrifugation. The insoluble fraction was resuspended in Tris 20 mM pH 8, filtered with a 0.22 μm filter and loaded into an anion exchange column HiTrap Q HP (GE Healthcare, Chicago, USA) coupled to an ÄKTA purifier high performance liquid chromatography system (GE Healthcare, Chicago, USA). Tris 20 mM pH 8 and Tris 20 mM pH 8, NaCl 1 M were used as buffer A and buffer B. αS was eluted using a step gradient: Step 1: 0%–20% buffer B, 5 cv; Step 2: 20%–45% buffer B, 11 cv; Step 3: 100% buffer B, 5 cv. Purified αS was dialyzed against 5 L ammonium acetate 50 mM in two steps; 4 h and overnight. Finally, protein purity was addressed using 15% SDS-PAGE. The purest fractions were lyophilized and stored at −80 °C. For the experiments, αS lyophilized aliquots were resuspended to a final concentration of 210 μM using PBS pH 7.4 and filtered using 0.22 μm filters. αS concentration was determined measuring the absorbance at 280 nm and using the extinction coefficient 5960 $M^{-1}$ $cm^{-1}$.

**Peptide preparation.** PSMα3, dPSMα3, All_Leu, All_Leu_19, Scaffold_19, Anionic Scaffold and LL-37 were purchased from Synpeptide (Shanghai, China) with a purity >95%. Single cysteine containing variants were purchased from Genscript (Piscataway, USA) with a purity >95%. LL-37 was diluted in Milli-Q sterilized water, divided into aliquots and lyophilized. Cysteine containing peptides were resuspended in PBS pH 7.4, 5 mM TCEP and subsequently labeled with the corresponding fluorophore. PSMα3, dPSMα3, All_Leu, All_Leu_19, Scaffold_19 and Anionic Scaffold were dissolved in a 1:1 mixture of trifluoroacetic acid and hexafluoroisopropanol and sonicated for 10 min. Stock solutions were divided into aliquots and vacuum dried with a SpeedVac (Thermo Fisher Scientific, Waltham, USA) and stored at −80 °C until assayed. Peptide aliquots were resuspended in pure Milli-Q water prior their use.

**αS and peptide labeling.** Site-specific labeling of αS was performed in an αS variant with a single engineered cysteine at position 122 (αS N122C). This variant was expressed and purified as described above but including 5 mM DTT in all purification steps. The protein was labeled with maleimide-modified Alexa Fluor 488 (AF488) (Invitrogen, Carlsbad, USA) for 15–20 h at 4 °C in the dark. After quenching the reaction with 10 mM DTT, free unreacted dye in the protein solution was subsequently separated using a P10 desalting column (GE Healthcare, Waukesha, USA), and the labeled protein solution was flash frozen with liquid nitrogen and stored at −80 °C. The different peptides, PSMα3, dPSMα3, and LL-37, were labeled at a single engineered cysteine at the N-terminus with maleimide-modified Atto647N (ATTO-TEC, Siegen, Germany). The same labeling and purification strategy were followed as for αS, although in this case the unreacted free dye was removed from the protein solution using a polyacrylamide desalting column (Thermo Fisher Scientific, Waltham, USA). Two cleaning steps were required to remove completely the free dye from the labeled peptide solution.

**Preparation of the different isolated αS aggregates samples.** For the isolation of type B* oligomers purified αS was dialyzed against Milli-Q water and lyophilized for 48 h in aliquots of 6 mg. The aliquots were resuspended in 500 μL of PBS pH 7.4 to a final concentration of ca. 800 μM, filtered through 0.22 μm filters and incubated at 37 °C without agitation for 20–24 h. The sample was then ultracentrifuged at 288,000 g in a SW55Ti Beckman rotor, in order to remove any possible fibrillar species formed during the incubation, and later filtered by four consecutive cycles of filtration through 100 kDa centrifuge filters (Merck, Darmstadt, Germany) in order to remove the great excess of monomeric protein from the oligomeric solution. Type A* oligomers were generated by incubating 210 μM of αS in PBS pH 7.4 with ten molar equivalents of (-)-epigallocatechin-3- gallate (EGCG) (Merck, Darmstadt, Germany) for 48 h at 37 °C. The excess of compound and unreacted monomeric protein were then removed by six consecutive cycles of filtration through 100 kDa centrifuge filters (Merck, Darmstadt, Germany). The concentration of the final oligomeric solutions was determined measuring the absorbance at 280 nm and using an extinction coefficient of 5960 $M^{-1}$ $cm^{-1}$ or absorbance at 495 nm and an extinction coefficient of 72,000 $M^{-1}$ $cm^{-1}$ for AF488-labeled oligomers. In all cases, the oligomers were kept at room temperature and were used within 3 days after their production. The fibrillar samples were produced as explained in the

aggregation kinetics methodology section. The non-reacted protein and small non-fibrillar species that could be formed during the aggregation reaction were removed from the sample by 3 consecutive steps of centrifugation and resuspension of the precipitated fraction in PBS buffer at pH 7.4. Fibrils were then sonicated 1 min, 50% cycles, 80% amplitude in a Vibra-Cell VC130 Ultrasonic Processor (Sonics, Newton, USA) to generate fibrillar samples with a relatively homogeneous size distribution of small fibrils. The concentration of the AF488-labeled fibrillar samples was determined by subtracting the absorbance of the monomer after centrifugation at 495 nm using an extinction coefficient of 72,000 $M^{-1}$ $cm^{-1}$, with respect to the total soluble protein at time 0. For type A* oligomers, the concentration was adjusted in situ for each experiment so that a suitable and consistent burst-rate was reached. Thus, an interference of EGCG in quantifying the sample was avoided.

**Far circular dichroism analysis.** Far-UV CD spectra of the different peptide solutions were recorded on a Jasco J-815 CD spectrometer (Halifax, Canada) Software- Jasco spectra manager v2 at 25 °C using samples of 15 μM peptide final concentration in Milli-Q water. CD signal was measured from 260 nm to 190 nm at 0.2 nm intervals, 1 nm bandwidth, 1 sec of response time and a scan speed of 100 nm/min on a 0.1 cm quartz cell. Ten accumulations were recorded and averaged for each measurement. For LL-37 peptide samples, CD spectra were recorded in PBS pH 7.4, because of structural differences of this peptide in water and saline solvents.

**Time-resolved fluorescence spectroscopy.** Dual-Color Time-Resolved Fluorescence Spectroscopy experiments were performed on a commercial MT200 (PicoQuant, Berlin, Germany) time-resolved fluorescence confocal microscope with a Time-Correlated Single Photon Counting (TCSPC) unit. Laser diode heads were used in Pulsed Interleaved Excitation (PIE), and the beams were coupled through a single-mode waveguide and adjusted to laser powers of 6 μW (481 nm) and 5 μW (637 nm) measured after the dichroic mirror for optimal count rates while avoiding photobleaching and saturation. The coverslip was placed directly on the immersion water on top of a Super Apochromat 60x NA 1.2 objective with a correction collar (Olympus Life Sciences, Waltham, USA). A dichroic mirror of 488/640 nm (Semrock, Lake Forest, IL, USA) was used as the main beam splitter. Out-of-focus emission light was blocked by a 50 μm pinhole and the in-focus emission light was then split by a 50/50 beamsplitter into two detection paths. Bandpass emission filters (Semrock, Lake Forest, IL, USA) of 520/35 for the green dye (AF488) and 690/70 for the red dye (Atto647N) were used before the detectors. Single Photon Avalanche Diodes (SPADs) (Micro Photon Devices, Bolzano, Italy) served as detectors. Each measurement had an acquisition time of 1–3 min.

For FCS experiments, the effective focal volume of the green channel and its structural parameters in our system were determined using a 1 nM solution of Atto488 (ATTO-TEC GmbH, Siegen, Germany) yielding $V_{eff, g}$ = 0.51 fL and $\kappa_g$ = 3.97. Positive and negative cross-correlation controls were performed with a dual-labeled dsDNA (10 nM) and an equimolar mixture (15 nM each) of AF488- and Atto647N-labeled monomeric αS (Supplementary Fig. 4). The positive control was also used for the determination of the red and dual-color effective focal volume and their structural parameter, yielding $V_{eff,r}$ = 0.1 fL, $V_{eff,gr}$ = 0.091 fL, $\kappa_r$ = 2.78, and $\kappa_{gr}$ = 2.67, respectively.

AF488-labeled aggregated αS samples were diluted in PBS pH 7.4 to a final protein concentration of ~1-5 nM in a 50 μL droplet which was spotted directly onto a cover glass (Corning, Corning, USA) previously coated with a 1 mg/mL BSA solution. Atto 647N-labeled peptides were titrated into the droplet and the peptide concentration was measured individually for each experiment by autocorrelation analysis of the red dye. No significant changes in correlation amplitudes were observed over time after equilibrating the samples for 2 min. Experiments were performed at 20 °C and samples were covered to avoid evaporation. It is important to note that, for αS aggregated species, consisting of several tens of monomers, the species concentrations are in the picomolar range and, as further explained below, single-particle conditions are ensured throughout the experiments. The aggregated species coexist with a certain amount of monomeric αS due to the stark sample dilution employed in the experiments and, therefore, the donor auto-correlation curves in Figs. 2, 6 and Supplementary Figs. 8, 12, 13, show both the diffusion component of the monomer and the aggregate. A similar behavior is observed for peptides PSMα3 and LL-37, which can exist as oligomerized species. For obtaining the diffusion coefficients of the different aggregates the diffusion component of the monomeric species in the samples was filtered out by intensity-filtered dcFCCS analysis as explained below. The diffusion coefficient ($D_g$ or $D_r$) fitted to data for αS species are 103 ± 16 $\mu m^2$ $s^{-1}$, 4 ± 0.9 $\mu m^2$ $s^{-1}$, 3.46 ± 1.2 $\mu m^2$ $s^{-1}$ and 0.81 ± 0.12 $\mu m^2$ $s^{-1}$ for the monomer, type A*oligomers, type B* oligomers and fibrils, respectively, in very good agreement with the diffusion coefficients expected according to their corresponding sizes as determined by AFM and DLS (see Supplementary Fig. 1) and as reported before[13,19]. In addition, such intensity thresholding yields a confocal volume mean occupancy ($N$) well below 1 for all fluorescent species involved, with $N$ = 0.019, $N$ = 0.043, and $N$ = 0.053 for type A*oligomers, type B* oligomers, and fibrils, respectively. Therefore, in terms of burst selection for the PIE-FRET and fluorescence stoichiometry analysis, where the same intensity threshold is applied, the experiments were conducted under single-particle conditions. This becomes even more evident when looking at the

raw data in the form of intensity time traces of, for instance, the PSMα3 - type B* oligomer interaction experiments (Supplementary Fig. 14).

$D_r$ values of $14.6 \pm 3.6$, $19.2 \pm 4.2$ μm$^2$ s$^{-1}$ and $108 \pm 18$ μm$^2$ s$^{-1}$ were calculated for PSMα3, LL-37, and dPSMα3, respectively, with $N = 0.053$ and $N = 0.11$, for PSMα3 and LL-37, respectively. These data indicate that PSMα3 and LL-37 exhibit a certain degree of oligomerization, despite no aggregates were detected in TEM images of peptides alone (not shown), and the data indicates that they bind to their targets in the monomeric form (Figs. 2, 6 and Supplementary Fig. 7).

Both data acquisition and analysis were performed on the commercially available software SymphoTime64 version 2.3 (PicoQuant, Berlin, Germany). For the oligomeric and fibrillar samples, a lower intensity threshold of 27 photons in the green dye autocorrelation analysis was applied to filter out the low intensity signal arising from the monomeric αS events generated upon dilution-induced disaggregation of the aggregated samples. This threshold was calculated as three times the mean intensity of monomeric αS obtained from the analysis of a sample of pure αS monomers. In addition, an upper intensity threshold was applied to auto-correlation and cross-correlation analysis to filter out any possible artifacts such as dust particles or aggregate clusters (even though these events were very scarce): 500 photons for monomer, type A* and type B* oligomers and 1500 photons for sonicated fibrils. Data on the red channel corresponding to the peptide fluorescence signal was intensity-filtered with a lower intensity threshold in analogy to the green channel owing to the fact that the peptide can also exist as self-assembled species. The reference signal was that of the monomer-only dPSMα3 sample. The PIE excitation scheme together with the TSCPC acquisition enabled the application of a lifetime-weighted filter, which aided removal of background and spectral cross-talk.

The corrected auto-correlations of the green and the red channel ($G_i$) were given by

$$G_i(\tau) = \frac{\langle F_i(t) \cdot F_i(t + \tau) \rangle}{F_i^2} - 1 \tag{1}$$

where $F_i(t)$ denotes the fluorescence intensity either the green or the red channel, $\tau$ is the correlation time and the angled brackets indicate a time average over the acquisition time. The cross-correlation ($G_x$) between the green and the red channel was given by

$$G_x(\tau) = \frac{\langle F_g(t) \cdot F_r(t + \tau) \rangle}{\langle F_g \rangle \langle F_r \rangle} - 1 \tag{2}$$

Auto-correlation curves for both the green and red channel were fitted with a 2 diffusion-component model accounting for residual monomeric αS and bound and unbound peptide, respectively, using the following equation:

$$G_i(\tau) = G_i^0 \frac{f_{i,1}}{\left(1 + \frac{\tau}{\tau_{D_{i,1}}}\right)\sqrt{1 + \frac{\tau}{\kappa^2 x \tau_{D_{i,1}}}}} + \frac{f_{i,2}}{\left(1 + \frac{\tau}{\tau_{D_{i,2}}}\right)\sqrt{1 + \frac{\tau}{\kappa^2 x \tau_{D_{i,2}}}}}, \tag{3}$$

where $G_i^0$ is the correlation amplitude at correlation time 0, $f_{i,1}$ and $f_{i,2}$ denote the fractional amplitudes of the monomeric and aggregated αS for the green channel (where i = g) and the bound and unbound peptide for the red channel (where i = r) and $\kappa^2$ is the structure parameter of the focal volume. The same applies for the diffusion terms $\tau_{D_{i,1}}$ and $\tau_{D_{i,2}}$. No correlated blinking is expected when multiple dyes are present on one particle as it is our case and therefore a blinking term was not included.

Cross-correlation amplitudes were fitted with a 1-component simple diffusion model since only one diffusion coefficient is expected for the interacting species (Supplementary Fig. 13) using the following equation:

$$G_x(\tau) = G_x^0 \frac{1}{\left(1 + \frac{\tau}{\tau_{D,x}}\right)\sqrt{1 + \frac{\tau}{\kappa^2 x \tau_{D,x}}}}. \tag{4}$$

With the corrected green dye autocorrelation function and the mean intensity of monomeric αS, the average aggregate particle number ($N_{Ag}$) for each αS aggregated sample was estimated as $N_{Ag} = \frac{1}{G_g^0}$. The average particle number for the peptide was calculated in analogy to that of αS. The peptide concentration was calculated as $C_P = \frac{N_r}{V_{eff,r} x N_A}$, where $N_r$ is the average number of particles in the red confocal volume, $V_{eff,r}$ is the red focal volume and $N_A$ is the Avogadro number. The cross-correlation amplitudes, dual-laser focal volume, $V_{eff,x}$, and peptide concentrations, $C_P$, were used for calculating the number of peptides bound to each αS species ($N_P$) and the free peptide concentration ($C_{P,Free}$) as described by Kruger and coworkers[32].

For single-burst FRET and stoichiometry analysis, an acceptor (red dye) direct excitation lower threshold based on the mean intensity of the time trace ($I_{A,mean} + 2 \times \sigma$) was used to filter out those events without an active acceptor molecule. To further select those events arising from αS aggregates, a burst selection intensity threshold of 100 photons was used. In the FRET analysis, experimentally determined correction factors were applied: spectral cross-talk $\alpha$ was 0.004, direct excitation $\beta$ was 0.0305 and detection efficiency $\gamma$ was 0.517. Burst-wise FRET

efficiency and stoichiometry were calculated as given by

$$E = \frac{F_{A,IE}}{F_D + F_{A,SE}} \tag{5}$$

$$S = \frac{F_D + F_{A,IE}}{F_D + F_{A,IE} + F_{A,DE}} \tag{6}$$

where $F_D$ is the fluorescence intensity in the donor (green) channel, $F_{A,IE}$ is the fluorescence intensity in the acceptor (red) channel through indirect excitation and $F_{A,DE}$ is the fluorescence intensity in the acceptor (red) channel after direct excitation by PIE pulse.

Stoichiometry values were corrected for the difference in mean intensity between the monomeric αS and peptide bursts, obtained from monomeric αS-only and peptide-only measurements; the obtained mean intensity ratio $I_{mean,αS}$: $I_{mean,peptide}$ was found to be 0.77. Stoichiometry distributions were fitted to a log-normal distribution to obtain the mean stoichiometry value for each measurement. The number of bound peptides per aggregate ($N_P$) was then estimated by multiplying the mean stoichiometry value previously obtained by the mean number of αS monomers present on each aggregate as calculated empirically from the molecular brightness in FCCS experiments. The free peptide concentration ($C_{P,Free}$) and $N_P$ obtained by either FCCS or single-burst stoichiometry analysis were used for calculating the binding curves as described by Kruger and coworkers[32]. To obtain the dissociation constant $K_D$ and the maximum specific binding sites $N_{max}$, the resulting binding curves were fitted to the following specific binding model with $n$ identical and independent binding sites:

$$Y = \frac{N_{max} \cdot X}{(K_D + X)} \tag{7}$$

The binding curves and binding parameters obtained from either FCCS or single-burst stoichiometry analysis were compared (Supplementary Fig. 7) and found to be remarkably similar, which validates the analysis. OriginPro9.1 software was used for graphical data representation and statistical analysis.

**Aggregation kinetics.** αS amyloid aggregation was monitored in a 96 wells plate (non-treated) (Sarstedt, Germany) containing Teflon polyballs (1/8″ diameter) (Polysciences Europe GmbH, Eppelheim, Germany) as described by Pujols and coworkers[9]. Each well contained 150 μL solutions of 70 μM αS in PBS buffer with 40 μM thioflavin-T and the corresponding concentration of peptide. Plates were incubated at 37 °C, 100 rpm in an orbital culture shaker Max-Q 4000 (Thermo Fisher Scientific, Waltham, USA). Aggregation was analyzed every 2 h using a Victor3.0 Multilabel Reader Software-PerkinElmer 2030. (PerkinElmer, Waltham, USA). End-point measurements were performed after 32 h of incubation. Fluorescence intensity was measured in triplicate by exciting with a 430–450 nm filter and collecting the emission with a 480–510 nm filter. The resulting kinetics were normalized to the maximum fluorescence of the αS control (untreated).

**Atomic force microscopy.** αS samples were diluted to a protein concentration of 0.1–0.5 μM and deposited on cleaved Muscovite Mica V-5 (Electron Microscopy Sciences; Hatfield, Pensilvania, USA). Slides were washed with double distilled water and allowed to dry before imaging acquisition on a Bruker Multimode 8 (Bruker; Billerica, USA) using a FMG01 gold probe (NT-MDT Spectrum Instruments Ltd., Russia) in intermittent-contact mode in air. Images were processed using Gwyddion (version 2.48) and the width measurements were corrected for the tip shape and size (10 nm).

**Polyacrylamide gel electrophoresis.** For monomeric and fibrillar αS species, 5 μg protein in denaturing loading buffer were loaded onto a 15% acrylamide SDS-PAGE. For type A* and type B*, 2 μg protein in non-denaturing buffer were loaded onto a 15% native-PAGE. The only difference between the denaturing and non-denaturing gel electrophoresis was the absence of SDS in the sample, gel and buffer of the native PAGE. No boiling step was included in either case. Unprocessed scans of the gels are presented in the Source Data file.

**Dynamic light scattering.** Estimations of the hydrodynamic radius of αS species were made on a DynaPro NanoStar (Wyatt, USA) equipped with a Peltier temperature control. Protein samples were prepared at a 25 μM concentration in filtered PBS (0.22 μm cellulose acetate syringe filters). DLS measurements were performed at 25 °C at a fixed angle of 90 °. Twenty acquisitions per measurement were collected using a 2 s acquisition time. An average of 10 measurements were performed for the statistical size analysis. Data was analyzed using the Dynamics software (version 6.12.03).

**Fourier-Transform infrared (FT-IR) spectroscopy.** αS aggregates species were transferred to deuterated buffer, by either centrifugation/resuspension or filtering cycles, to a final protein concentration of ca. 4 mg/ml. Samples were then deposited between two CaF2 polished windows separated by a PTFE Spacer (Harrick Scientific Products Inc., USA). Spectra were collected in transmission mode at room temperature using a VERTEX 70 FTIR Spectrometer (Bruker, USA) equipped with

a cryogenic MCT detector cooled in liquid nitrogen. IR spectra were processed and analyzed using standard routines in OPUS version 6.5 (Bruker, USA), RAMOPN (NRC, National Research Council of Canada) and Spectra-Calc-Arithmetic© version A2.21 (Galactic Inc., USA).

**ANS fluorescence spectroscopy**. 10 μM of each αS sample was incubated with 500 μM 8-anilo-1-naphtalene-sulfonic acid (ANS) in PBS for 45 min before recording the spectra. The extinction coefficient of ANS at 350 nm was assumed to be 5000 $cm^{-1} M^{-1}$. In order to monitor ANS binding to the each αS species, samples were excited at 350 nm and their emission spectra were recorded from 400 to 650 nm in 1-nm steps. Spectra were collected at room temperature in a Cary Eclipse Fluorescence Spectrophotometer (Varian, Palo Alto, California, United States) with slit-widths of 5/5 nm. An averaging time of 100 ms was used.

**Transmission electron microscopy**. For electron microscopy analyses, end-point aggregated samples were sonicated for 5 min at minimum intensity in an ultrasonic bath (VWR ultrasonic cleaner) and placed onto carbon-coated copper grids and allowed to adsorb for 5 min. The grids were then washed with distilled water and negative stained with 2% (w/v) uranyl acetate for 1 min. Finally, the excess of uranyl acetate was absorbed using ashless filter paper and the grids were left to air-dry for 15 min. A TEM JEM-1400 Software-Gatan Digital Micrograph 1.8 (JEOL, Peabody, USA) microscope was used operating at an accelerating voltage of 120 kV. The more representative images of each grid were selected. Images were processed and analyzed with Image J (version 1.52p)

**Sedimentation assay**. αS aggregation was performed as previously described. End-point samples were subjected to ultracentrifugation at 100,000 g for 30 min at 20 °C in a SW55Ti Beckman rotor in order to fractionate soluble and fibrillar species. αS concentration in the soluble fraction was determined by measuring the absorbance at 280 nm (ε = 5960 $M^{-1} cm^{-1}$). Soluble fractions in denaturing loading buffer were boiled for 5 min and loaded onto a 15% acrylamide SDS-PAGE. Proteins were revealed with BlueSafe (NZYTech, Portugal). Unprocessed scans of the gels are presented in the Source Data file.

**Isolation of low molecular weight aggregates generated during αS in vitro aggregation**. αS aggregation was performed as previously described in absence and presence of PSMα3. Aliquots at the analyzed time point were taken and flash frozen in liquid nitrogen and stored at −80 °C until assayed. To fractionate our sample into insoluble species, low-molecular weight aggregates and monomers, we adapted the centrifugation-based protocol developed by Kumar and coworkers[51]. αS preparations were subjected to ultracentrifugation at 100,000 g for 30 min at 20 °C in a SW55Ti Beckman rotor in order to isolate larger fibrillar species. The soluble fraction (100 μl) containing low molecular weight aggregates and monomeric αS was then filtrated through 100 kDa centrifuge filters (Merck, Darmstadt, Germany) in order to fractionate these two species. The filtrated samples contain monomeric or -theoretically- dimeric αS. The excess of monomeric species retained in the filter were then washed by filtrating 400 μl of PBS. Finally, aggregated species retained in the filter were recovered by adding 100 μl of PBS to the membrane and carefully pipetting. This fraction containing low molecular weight aggregates was subsequently analyzed by transmission electron microscopy as previously described above.

**Neuroblastoma culture**. Human SH-SY5Y neuroblastoma cells (ATCC) were cultured in DMEM/F12 medium supplemented with 15% FBS and 1xNEAA. Cells were grown at 37 °C in a 5% $CO_2$ humidified atmosphere until an 80% confluence for a maximum of 20 passages.

**Analysis of intracellular ROS**. SH-SY5Y cells were seeded onto glass coverslips (Ibidi, Gräfelfing, Germany) at $0.5 \times 10^6$ cells/mL and treated for 15 min with 10 μM of type B* oligomers or type B* pretreated for 15 min with the tested peptide (PSMα3, dPSMα3, and LL-37). Then, CellROX® Green (Invitrogen, Carlsbad, USA) at a final concentration of 5 μM was added and incubated for 30 min at 37 °C. Cells were washed with PBS and fixed with 3.7% paraformaldehyde (PFA) for 15 min. The intracellular fluorescence of the SH-SY5Y cells was analyzed on a Leica TCS SP5 Software-Gatan Digital Micrograph 1.8 (Leica Microsystems, Wetzlar, Germany) with a HCX PL APO 63 × 1.4 oil immersion objective, under UV light by using a 488 nm excitation laser for CellROX and collecting the emission with a 515–560 nm filter range. Images were processed and analyzed with Image J (version 1.52p)

**Oligomer binding to cells**. SH-SY5Y cells were seeded onto glass coverslips (Ibidi, Gräfelfing, Germany) and treated for 45 min with 10 μM of type B* oligomers or type B* pretreated for 15 min with an equimolar concentration of PSMα3 or dPSMα3. Cells were then washed with PBS and fixed with 3.7% PFA for 15 min. Then cells were washed with PBS containing 0.1% Triton X-100 for 10 min. Cells were blocked with 5% BSA-PBS and incubated with 1/200 dilution rabbit polyclonal anti-αS antibody (Abcam, Cambridge, UK) overnight at 4 °C, and with 1:1000 anti-rabbit secondary antibodies conjugated with AF488. Cell nuclei was

stained using Hoescht 33342 at a concentration of 0.5 μg/mL for 5 min. Images of intracellular αS were obtained under UV light using double excitation at 488 nm and 350 nm lasers, for AF488 and Hoescht, and the emission was collected at 515–560 nm and 405 nm, respectively. Images were processed and analyzed with Image J (version 1.52p)

A dot blot assay was performed as a control to discard epitope-masking artifacts caused by the potential primary or secondary antibodies binding to the peptide. 2 μL of type B* oligomers untreated and treated for 15 min with an equimolar concentration of PSMα3 were spotted onto a nitrocellulose membrane and allowed to dry. Antibody incubations were performed as described for the cellular assay. No significant differences in the signals of oligomers assayed in the presence or the absence of PSMα3 were detected (data not shown).

**Redesign of PSMα3 variants**. To guide and assist the design of PSMα3 peptide variants some computational tools were employed. Briefly, AGADIR was used to predict the helical propensity of the peptide variants based on the helix/coil transition theory[26]. FoldX allows a rapid evaluation of the effect of mutations on the stability, folding and dynamics of proteins[38]. We exploited it to evaluate if the designed mutations may compromise the stability of the α-helix specially regarding extensive redesign or those involving electrostatic repulsions. The peptides mean hydrophobicity ($H$), and their helical hydrophobic moment ($\mu_H$), a measure of the amphiphilicity of a helix, were calculated according to Eisenberg and coworkers[52].

**Reporting summary**. Further information on research design is available in the Nature Research Reporting Summary linked to this article.

## Data availability

All the data presented in this study are available in the paper or in the Supplementary Information. Further raw data (i.e., Time traces of the time-resolved fluorescence spectroscopy) supporting the findings of this study are available from the corresponding author upon reasonable request. The screened database of human peptides was obtained from the EROP-Moscow oligopeptide database (http://erop.inbi.ras.ru/)[39]. Source data are provided with this paper.

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

## Acknowledgements

This work was funded by the Spanish Ministry of Economy and Competitiveness (MINECO) BIO2016-78310-R and BIO2017-91475-EXP to S.V, by the Ministry of Science and Innovation (MICINN) PID2019-105017RB-I00 to S.V, by ICREA, ICREA-Academia 2015 to S.V. N.C. was supported by MINECO RYC-2012-12068, MINECO/FEDER, EU BFU2015-64119-P and MICIU/FEDER, EU PGC2018-096335-B-100. J.S. was supported by MICINN via a doctoral grant (FPU17/01157). We thank the members of the Microscopy Services of the UAB for their assistance, Dr. Salvador Bartolomé from the Laboratori de Luminiscència i Espectroscòpia de Biomolècules, UAB, and Dr. Evangelos Sisamakis, PicoQuant, for insightful discussions.

## Author contributions

S.V. conceived the project. J.S., I.P., and S.V. designed all the peptide sequences. J.S., S.P-D., and J.P. performed in vitro experiments. J.S. and S.N. designed, conducted and analyzed cellular experiments. P.G. and N.C. performed and analyzed time-resolved fluorescence spectroscopy experiments. J.S., I.P., and S.V. analyzed the data. J.S., P.G., N.C., I.P., and S.V. prepared the paper with contributions from all the authors.

## Competing interests

Authors have submitted a patent application on the bases of the here presented research. It is protected the use of LL-37 for therapy and PSMa3 for therapy and diagnosis. Inventors: S.V., I.P., J.S., N.C., P.J.G. Title: inhibitors of alpha synuclein aggregation and uses thereof. Property: Universidad Autónoma de Barcelona. Universidad de Zaragoza. Request number: EP20382658. Priority date and Country: 22-07-2020, EU. All other authors declare no competing interests.

## Additional information

**Peer review information** *nature communications* thanks Hilal Lashuel and other, anonymous, reviewers for their contributions to the peer review of this work. Peer review reports are available.

