## [Peer Review File · Nature Communications]

Editorial Note: Parts of this Peer Review File have been redacted as indicated to remove third-party material where no permission to publish were obtained.

Reviewers' comments:

Reviewer #1 (Remarks to the Author):

§In the manuscript submitted by Santos et al., the authors present a family of α -helical peptides targeting α -synuclein (α -syn), the protein associated with Parkinson's disease and other neurodegenerative disorders. In particular, the α -helical peptides are able to bind with low nanomolar affinity to oligomers that the authors have previously shown to be the most toxic forms of α -syn. To demonstrate this, the authors made use of dual-color fluorescence cross-correlation spectroscopy (dcFCCS) to measure the affinity between a labeled bacterial extracellular peptide and various labeled conformers of α -syn. They found that although the affinity of the peptide to the different α -syn aggregates was similar, there were a greater number of binding sites on the Type B oligomers and fibrils than on the Type A oligomers. They confirmed this binding with pulsed interleaved excitation (TCSPEC) measurements. They further went on to show that the peptide was able to inhibit the aggregation of α -syn, and could protect cells from α -syn oligomer-induced toxicity. Based on the structural characteristics of the bacterial peptide, one expressed in humans was identified, and also shown to bind to α -syn oligomers, reducing their toxicity.

The work outlined in the manuscript is of a high quality, and will be of great interest to those familiar with α -syn. Whilst I am content with most of the findings, there are a few points that should be clarified before the manuscript is accepted.

1) Specificity of the peptide to α -syn species:

The claim is made that the peptides are α -syn species-specific, and whilst the authors demonstrate that they only bind certain α -syn aggregates, there is no evidence to suggest specificity to α -syn. Indeed, they note that LL-37 is also able to bind A β -42 and IAPP monomers. More careful phrasing could be used to ensure that the readership is not led into believing that the peptides only bind α -syn aggregates.

2) Diffusivity of the peptide:

The FCS traces are presented in Figure 2, and whilst the cross-correlation does show binding between the peptide and the α -syn aggregates, the characteristic diffusion time, τ_D , is higher than I would expect for a 22 AA peptide. From Figure 2A, $\tau_D \sim 0.2$ ms for α -syn and ~ 1 ms for the peptide. Taking into account the confocal volumes and κ values from the materials and methods, a diffusion coefficient of $\sim 100 \mu\text{m}^2 \text{s}^{-1}$ can be calculated for monomeric α -syn (similar to that found in Nath et al. Biophys J., 2010); however, for the peptide, a value of $\sim 8 \mu\text{m}^2 \text{s}^{-1}$ is calculated. This is much lower than expected, and using the Stokes' Einstein relation, would lead to a hydrodynamic radius of ~ 25 nm. Can the authors explain this? Could it be that the peptide is itself aggregated, as has been shown in Marinelli et al., Sci. Rep. 2016 amongst other publications? The peptide in the negative control has a diffusion coefficient more as I would expect.

As the data are present, it would be of benefit to the reader to include all of the diffusion coefficients for the different aggregates and peptides.

3) Pulsed Interleaved Excitation (PIE) experiments:

In the materials and methods section, it appears that the PIE experiments were performed at a concentration of ~ 5 nM. This seems quite high for single-molecule detection, and if accurate stoichiometries are to be calculated, the mean occupancy of the confocal volume must be kept below 1. Was this the case? Taking into account the confocal volume, the occupancy would be greater than 1.

Also, the number of events detected seems rather high for a detection time of only 2 minutes of data acquisition.

The authors show FRET histograms in the SI, but do not show either the stoichiometry histograms, or 2D plots of FRET and Stoichiometry. These would arguably be more useful, particularly as these are used to calculate the binding curve in SI Fig 9. Can these fitted stoichiometry histograms be included?

4) Cell experiments:

The claim is made that the oligomers bind to the cells; however, a more accurate description would be that the oligomers are internalised. Can the authors explain why they used unlabelled α -syn followed by fixation/permeabilization and immunohistochemistry, rather than taking advantage of the labelled α -syn to directly visualize entry (and inhibition thereof) into cells? This could also be done with live-cell imaging.

Could it be the case that α -syn is still internalised in the presence of the peptide, but that the epitope is not accessible to the antibody due to the bound peptide? A control experiment could be done to show that this isn't the case, for example, imaging the aggregates in the absence of cells using the antibodies.

Would it also be possible to determine whether the peptide is able to rescue cells that already have oligomers internalised?

Also, control experiments with the disordered peptide should be performed to show that there is no effect from the peptides themselves.

Minor comments:

I assume that the peptide that is referred to in the manuscript as PMSa3 is in fact PSMa3?

Reviewer #2 (Remarks to the Author):

The process of alpha-synuclein protein aggregation has been the subject of numerous studies over the past 10 years, especially due to the relevant role it plays in the development of Parkinson disease. Unfortunately, despite the efforts of numerous research groups, we do not yet have any compound today that in a controlled manner has an impact on protein aggregation potent and specific enough to translate into a therapeutic effect. In this context, the study presented by Santos et al. represents a major breakthrough. First, the authors describe compounds capable of interacting and preventing the aggregation of non-toxic oligomers that are formed in the early stages of the process. These synthetic peptides, moreover, do not interact with the monomer, and this is of great importance so as not to interfere with the physiological role of the protein. Secondly, already from a methodological point of view, this article illustrates the power of molecular design methods developed in recent years in Ventura's laboratory. It is fantastic to realize how with seemingly simple rules, peptides capable of performing complex molecular recognition tasks can be designed. The article is well written, the bibliographic references are very complete, and the experimental work is described in a rigorous way. For all this, my recommendation is that the article be accepted for publication in Nature Communications.

Reviewer #3 (Remarks to the Author):

Santos J et al., describe studies aimed at the development of α -helical peptides that selectively target alpha-synuclein oligomers and fibrils. They then suggest that these peptides exhibit potent anti-aggregation activity and inhibit alpha-synuclein oligomer toxicity. They then report on the identification of a human peptide that also inhibits alpha-synuclein oligomerization and toxicity. Unfortunately, the lack of data on the characterization of the aSyn preparations they used in this study and the mode of action of these peptides makes it difficult to evaluate their interpretations of the results and the main claims of the paper.

The authors present this work as a rational design of alpha-synuclein inhibitors, but then do not show how the design of the peptide was guided by the structural insights from our current understanding of alpha synuclein oligomers and their diversity. Instead, they simply say that they identified the PMSa3 and that it fulfilled the desired criteria.

The entire manuscript is focused on designing specific helical peptide inhibitors based on exploiting the structural properties of four particular types of alpha-synuclein oligomers (A/A* and B/B*) and fibrils that the authors claim to possess distinct conformation, morphology, size, hydrophobicity, and toxic properties. Yet, nowhere in the manuscript do the authors present data on the biochemical and biophysical homogeneity/heterogeneity of the oligomers and their conformational properties. Simply citing the previous publications, some of which do not provide the information mentioned, is not sufficient and does not allow one to properly assess the experiments and the interpretation of the results from the experiments performed by the authors.

The authors claim on several occasions that they investigated the interactions of the peptides with four major alpha synuclein species, implying that they are dealing with homogeneous preparations of alpha synuclein oligomers and fibrils. However, all the methods used by the others to generate oligomers are known to yield heterogeneous mixtures of oligomers rather than a specific type of oligomers. The methods used here, repeated concentration procedures, and centrifugation through 100 KDa, could also alter the structure and distribution of the oligomers in the samples.

Therefore, it is imperative that the authors provide the following data on their oligomer and fibril preparations.

- 1) Electron microscopy data (Multiple EM images) that allow for assessment of the morphology of the oligomers in each preparation
- 2) Light scattering or sedimentation velocity to assess the size distribution of the oligomers.
- 3) Electron microscopy data on the fibril preparations
- 4) SDS-PAGE analysis of each preparation to establish their purity and the absence of truncated alpha synuclein species.
- 5) Amounts of monomers in each preparation

The design rationale is on the basis that the authors have obtained structural data on individual types of oligomers, which is not possible given the heterogeneity of the samples. They do not present any data to demonstrate that the oligomers they used possess the design features they used to target (hydrophobic patches embedded in the anionic environment) or present data that demonstrate how these features are different in different types of oligomers.

The authors also indicate that the oligomers were kept at room temperature for three days. It is not

clear why they did this, but what is important is to establish that the biophysical properties did not change during this incubation period. Usually, oligomer preparations are stored at -20C or 4C.

The authors should present data on the oligomerization states of the peptides. They should establish using robust biophysical methods the oligomerization state of these peptides and whether they bind to alpha synuclein as monomers or oligomers.

The authors do not present any data that establish the mode of binding to different aSyn species and which regions of alpha synuclein interact with these peptides.

The authors do not provide any data that establishes the exact mode of action for the peptide inhibitors or show that they indeed act by preventing oligomer to fibril transition, as they claim in the paper (page 9). They rely mainly on ThT signal intensity of EM, both of which are not quantitative. To show that these peptides indeed target oligomers and assess which species along the pathway of aggregation they target, they need to use more quantitative techniques to determine the distribution of the different alpha synuclein species in the presence and absence of peptide inhibitors. This can be achieved easily using simple sedimentation- or SEC-based techniques. If their proposed mechanisms of action are correct, then they should be able to see the accumulation of alpha synuclein oligomers-peptide complexes. Also, the EM data for all the peptides should be included as supporting information.

It is puzzling that the authors use the same exact ThT data for aSyn in all the figures despite the fact that the various experiments were performed in different periods. One would expect that they included alpha synuclein alone samples in each aggregation experiments and the effect of the inhibitors should be compared to the untreated alpha synuclein samples in the same experiment and in the same plate. They should explain and justify this.

The authors claim that they used the absorbance at 280 nm to determine the concentration of the oligomers. However, this method may not be accurate for the oligomers prepared by co-incubation of alpha-synuclein with EGCG.

There is no point of comparing the activity of the peptide inhibitors here to other inhibitors unless they were assessed under the same conditions and in the same assays. Therefore, the comparison to SynuClean-D and other molecules (Anle138b) here is not valid or fair.

The ability of the peptide to abolish alpha synuclein-oligomers-induced increase ROS is not consistent with the fact that it only inhibited oligomer membrane binding and uptake by only ~60%.

The authors suggest that the peptide LL-37 exhibit similar tissue distribution as alpha synuclein without showing any data to support their claim. They should elaborate on this and present the data.

Reviewers' comments:

Reviewer#1 (Remarks to the Author):

In the manuscript submitted by Santos et al., the authors present a family of α -helical peptides targeting α -synuclein (α -syn), the protein associated with Parkinson's disease and other neurodegenerative disorders. In particular, the α -helical peptides are able to bind with low nanomolar affinity to oligomers that the authors have previously shown to be the most toxic forms of α -syn. To demonstrate this, the authors made use of dual-color fluorescence cross-correlation spectroscopy (dcFCCS) to measure the affinity between a labeled bacterial extracellular peptide and various labeled conformers of α -syn. They found that although the affinity of the peptide to the different α -syn aggregates was similar, there were a greater number of binding sites on the Type B oligomers and fibrils than on the Type A oligomers. They confirmed this binding with pulsed interleaved excitation (TCSPC) measurements. They further went on to show that the peptide was able to inhibit the aggregation of α -syn, and could protect cells from α -syn oligomer-induced toxicity. Based on the structural characteristics of the bacterial peptide, one expressed in humans was identified, and also shown to bind to α -syn oligomers, reducing their toxicity.

The work outlined in the manuscript is of a high quality, and will be of great interest to those familiar with α -syn. Whilst I am content with most of the findings, there are a few points that should be clarified before the manuscript is accepted.

1) Specificity of the peptide to α -syn species:

The claim is made that the peptides are α -syn species-specific, and whilst the authors demonstrate that they only bind certain α -syn aggregates, there is no evidence to suggest specificity to α -syn. Indeed, they note that LL-37 is also able to bind A β -42 and IAPP monomers. More careful phrasing could be used to ensure that the readership is not led into believing that the peptides only bind α -syn aggregates.

We fully agree with the reviewer, and accordingly, a more precise phrasing has been used to clarify this issue in the newly submitted version of the manuscript.

2) Diffusivity of the peptide:

The FCS traces are presented in Figure 2, and whilst the cross-correlation does show binding between the peptide and the α -syn aggregates, the characteristic diffusion time, τ_D , is higher than I would expect for a 22 AA peptide. From Figure 2A, $\tau_D \sim 0.2$ ms for α -syn and ~ 1 ms for the peptide. Taking into account the confocal volumes and κ values from the materials and methods, a diffusion coefficient of $\sim 100 \mu\text{m}^2 \text{s}^{-1}$ can be calculated for monomeric α -syn (similar to that found

in Nath et al. Biophys J., 2010); however, for the peptide, a value of $\sim 8 \mu\text{m}^2 \text{s}^{-1}$ is calculated. This is much lower than expected, and using the Stokes' Einstein relation, would lead to a hydrodynamic radius of $\sim 25 \text{ nm}$. Can the authors explain this? Could it be that the peptide is itself aggregated, as has been shown in Marinelli et al., Sci. Rep. 2016 amongst other publications? The peptide in the negative control has a diffusion coefficient more as I would expect. As the data are present, it would be of benefit to the reader to include all of the diffusion coefficients for the different aggregates and peptides.

The reviewer's observation is correct; the helical peptides present a certain degree of self-assembly that results in a slower diffusion than expected for a monomer ($19 \mu\text{m}^2 \text{s}^{-1}$). Instead, the peptide used as a negative control (non-helical peptide dPSM3) shows a faster diffusion, compatible with most molecules diffusing as single peptides ($108 \mu\text{m}^2 \text{s}^{-1}$). Therefore, we assume that the oligomerization of the helical peptides is guided by the amphipathic nature of the helices, which expose a highly hydrophobic face to the solvent and, therefore, might be prone to establish intermolecular interactions.

Even though, as the reviewer points out, the peptide can suffer oligomerization due to its amphipathic nature, we have observed in our experiments that the monomeric form of the peptide can effectively bind the αS aggregated species. Specifically, our dcFCCS-derived binding curves in **Figure 2** and **Figure 6** indicates that monomeric peptide is bound to the aggregates because N_p values as low as 1 are observed for type A* and type B* oligomers and fibrils. The same is true for the binding curves and the N_p values obtained by single-particle fluorescence stoichiometry analysis, as shown in **Supplementary Figure 7**.

We have clarified these points in the revised version of the manuscript; see lines 210-214 in the main text and supplementary information of the revised version of the manuscript. Additionally, all the diffusion coefficients of the aggregates and peptides have being included in lines 136-156 of the supplementary information.

3) Pulsed Interleaved Excitation (PIE) experiments:

In the materials and methods section, it appears that the PIE experiments were performed at a concentration of $\sim 5 \text{ nM}$. This seems quite high for single-molecule detection, and if accurate stoichiometries are to be calculated, the mean occupancy of the confocal volume must be kept below 1. Was this the case? Taking into account the confocal volume, the occupancy would be greater than 1.

For the sake of clarity, we presented all concentrations of dcFCCS as well as dual-color single-particle fluorescence spectroscopy experiments in mass concentration, to reflect the concentration of αS molecule units composing each αS aggregate sample, as, in principle, each peptide molecule might interact with each αS molecule within the αS aggregates.

However, it must be noted that in the samples containing protein aggregates and peptide/aggregate complexes, the number of protein and peptide species is drastically reduced as compared to the number of α S and peptide molecule units in the sample. For example, the α S species concentrations in the α S aggregate samples are at least one order of magnitude smaller than the reported mass concentration, as the aggregates have more than 10 α -S molecule units per aggregate (indeed, ca. 30 protein units for type A* and type B* oligomers and more than 50 protein units for fibrils according to literature (Fusco G. et al., **Cremades N**, Ying L, Dobson CM, De Simone A. Structural basis of membrane disruption and cellular toxicity by alpha-synuclein oligomers. *Science* (2017) 358:1440; Chen SW et al., and **Cremades N**. Structural characterization of toxic oligomers that are kinetically trapped during alpha-synuclein fibril formation. *PNAS USA* (2015) 112: E1994), and our own analysis based on the fluorescence molecular brightness differences between the monomer and the aggregated species, see page 8, lines 191-193 of the revised version the manuscript).

As a matter of fact, the dcFCCS intensity thresholding described in the “Methods” section yields a confocal volume occupancy, $\langle N \rangle$, of 0.019, 0.043, and 0.027 for type A* oligomers, type B* oligomers and fibrils, respectively (volume occupancies below 1 are also obtained for the helical peptides given its self-assembled nature). This is well below 1 and therefore shows that, in terms of burst selection for the PIE-stoichiometry analysis, where the same intensity threshold is applied, the experiments were conducted under single-particle conditions and the burst-wise analysis is thus valid.

We have now clarified this point, lines 162-165 of the revised version of the manuscript, lines 128-149 in the supplementary information and new **Supplementary Figure 13**.

Also, the number of events detected seems rather high for a detection time of only 2 minutes of data acquisition.

In the revised version of the manuscript, the newly provided **Supplementary Figure 13** shows representative 1-second time-traces of binding experiments with type B* oligomers PSM α 3 as an example. In these time-traces, a number that usually is between 2 and 8 interacting particles is observed. For 3-minute data acquisitions, we believe that values around 1000 events are reasonable. The same is observed for other binding measurements.

The authors show FRET histograms in the SI, but do not show either the stoichiometry histograms, or 2D plots of FRET and Stoichiometry. These would arguably be more useful, particularly as these are used to calculate the binding curve in SI Fig 9. Can these fitted stoichiometry histograms be included?

We appreciate that the reviewer raised this point. We agree that showing the stoichiometry histograms would significantly help the reader follow the analysis and conclusions of this part of

the article. Some of the fitted histograms have been now included in the revised version of the manuscript; **Supplementary Figure 6**.

4) Cell experiments:

The claim is made that the oligomers bind to the cells; however, a more accurate description would be that the oligomers are internalised.

This is a good observation. We, on purpose, used the terms binding/ α S load per cell to account for all the oligomers interacting with cells, either internalized or inserted in the membrane. In our experiment's timeframe, a significant fraction of the oligomers remains inserted into the cell membrane but are not internalized. Since membrane perturbation is a reported oligomer toxicity mechanism, we intentionally account for both internalized and inserted species.

Can the authors explain why they used unlabelled α -syn followed by fixation/permeabilization and immunohistochemistry, rather than taking advantage of the labelled α -syn to directly visualize entry (and inhibition thereof) into cells? This could also be done with live-cell imaging.

The cell experiments were performed with unlabeled α S since we decided to follow a previously described and widely used immunohistochemistry protocol for the analysis of oligomer interaction with cells (Perni M et al. Multistep Inhibition of α -Synuclein Aggregation and Toxicity in Vitro and in Vivo by Trodusquemine. **ACS Chem Biol** (2018)). Using labelled oligomers in a cellular experiment would be significantly more time consuming and expensive (the yield of preparing labeled oligomers is only around 1%), and we considered that immunohistochemistry would be a powerful enough approach to assay how the interaction of the peptides with the toxic oligomers impacted the cells.

Could it be the case that α -syn is still internalized in the presence of the peptide, but that the epitope is not accessible to the antibody due to the bound peptide? A control experiment could be done to show that this isn't the case, for example, imaging the aggregates in the absence of cells using the antibodies.

This is a pertinent observation. Accordingly, we have performed a dot blot assay to verify the antibody's performance and discard any potential epitope-masking artifact. No appreciable differences in the signal were recorded when the oligomers were assayed in the presence or in the absence of peptide (**Figure R1**). This result is consistent with the polyclonal nature of the primary antibody. Thus, the dot blot result demonstrates that the peptide does not interfere with the antibody binding to the oligomers. The absence of peptide interference in antibody detection is now included in the "Methods" section (lines 344-349 in the revised version of the supplementary information).

Figure R1: Dot blot assay. 2 μ L of α S oligomers untreated or pretreated with an equimolar concentration of PSM α 3 were dotted in a nitrocellulose membrane. Antibody detection was tested as described for the cellular assay. No significant differences in the signal were detected in the absence and presence of PSM α 3.

Would it also be possible to determine whether the peptide is able to rescue cells that already have oligomers internalised?

We thank the reviewer for this suggestion. Theoretically, a significant amount of cell damage comes from a perturbation of membranes in a relatively short timeframe. Thus, it is not expected that our peptides will reverse this effect once oligomers are inserted into the membrane.

Also, control experiments with the disordered peptide should be performed to show that there is no effect from the peptides themselves.

We fully agree with the reviewer, and this control experiment has been now performed and included in the new version of the manuscript (**Figure 4**). As expected, no differences were observed in the amount of α S bound to cells when treated with the disordered peptide, relative to untreated oligomers. Thus, this control experiment indicates the peptide itself does not affect the binding of the oligomers to cellular membranes. The result of this control experiment is included in the new version of the manuscript (lines 281-282 in the main text), together with a new panel in Figure 4 and its description in the "Methods" section.

Minor comments:

I assume that the peptide that is referred to in the manuscript as PMS α 3 is in fact PSM α 3?

We thank the reviewer for spotting this typo; it is now corrected in the revised version of the manuscript.

Reviewer #2 (Remarks to the Author):

The process of alpha-synuclein protein aggregation has been the subject of numerous studies over

the past 10 years, especially due to the relevant role it plays in the development of parking disease. Unfortunately, despite the efforts of numerous research groups, we do not yet have any compound today that in a controlled manner has an impact on protein aggregation potent and specific enough to translate into a therapeutic effect. In this context, the study presented by Santos et al. represents a major breakthrough. First, the authors describe compounds capable of interacting and preventing the aggregation of non-toxic oligomers that are formed in the early stages of the process. These synthetic peptides, moreover, do not interact with the monomer, and this is of great importance so as not to interfere with the physiological role of the protein. Secondly, already from a methodological point of view, this article illustrates the power of molecular design methods developed in recent years in Ventura's laboratory. It is fantastic to realize how with seemingly simple rules, peptides capable of performing complex molecular recognition tasks can be designed. The article is well written, the bibliographic references are very complete, and the experimental work is described in a rigorous way. For all this, my recommendation is that the article be accepted for publication in Nature Communications.

We thank the reviewer for her/his positive comments on our work.

Reviewer #3 (Remarks to the Author):

Santos J et al., describe studies aimed at the development of α -helical peptides that selectively target alpha-synuclein oligomers and fibrils. They then suggest that these peptides exhibit potent anti-aggregation activity and inhibit alpha-synuclein oligomer toxicity. They then report on the identification of a human peptide that also inhibits alpha-synuclein oligomerization and toxicity. Unfortunately, the lack of data on the characterization of the aSyn preparations they used in this study and the mode of action of these peptides makes it difficult to evaluate their interpretations of the results and the main claims of the paper.

We thank the reviewer for his/her thoughtful review and dedication to our work. We can understand the concerns presented in the revision, especially considering the heterogeneity typically associated with protein aggregation. About this first concern, we consider that the methods employed for preparation and isolation of the different α S oligomeric samples are well-established procedures in the field and the resulting preparations have been extensively characterized in recent years; particularly in:

- Chen SW, et al., Structural characterization of toxic oligomers that are kinetically trapped during alpha-synuclein fibril formation. *PNAS* (2015). 112:E1994-E2003 (doi: 10.1073/pnas.1421204112) (morphological, structural and stability characterization of Type B* oligomers)
- Fusco G, et al., Structural basis of membrane disruption and cellular toxicity by alpha-synuclein oligomers. *Science* (2017) 358:144-30 (doi: 10.1126/science.aan6160) (characterization and structure determination of Type A* and B* oligomers and molecular origins of toxicity of Type B* oligomers)

- Chen SW and Cremades N. Preparation of alpha-synuclein amyloid assemblies for toxicity experiments. *Methods Mol. Biol.* (2018) 1779:45-60 (doi: 10.1007/978-1-4939-7816-8_4) (protocol for the preparation of stable and highly homogeneous samples of Type B* oligomers and short fibrils)

We have been directly involved in the characterization of these α S preparations through the different publications, and have already reported the reproducibility, purity and stability of both types of oligomeric samples (please, see the figures at the end of the text). Indeed, the quality and homogeneity of the oligomeric samples have allowed their structural determination by means of cryo-EM and solid-state NMR analysis, which require a high degree of sample structural homogeneity.

To illustrate the main morphological and structural features of the different α S species in the different preparations used in these study, in the revised version of the manuscript, we have included a Supplementary Figure (**Supplementary Figure 1**), with the analysis of the size, morphology, purity, structure and hydrophobicity of the different α S species using a wide range of biophysical techniques. The results obtained are identical to those we have previously published.

As for the lack of data on the peptide mode of action, we firmly believe that such data was already included in the manuscript. Additionally, prompted by the reviewer's suggestion, we provide new insights into the mechanism of action by which PSM α 3 inhibits α S amyloid formation *in vitro*. By analyzing the low-molecular species generated at the early stages of aggregation -in the absence and presence of peptide- we found that PSM α 3 blocks or at least delays the progression of annular oligomers into fibrils (new **Figure 3d** and **Supplementary Figure 10**). Annular oligomers morphologically resemble type B* oligomers (see above references) and oligomeric species previously identified in α S aggregation reactions:

- Lashuel, H. A., et al. Neurodegenerative disease: amyloid pores from pathogenic mutations. *Nature* (2002), 418, 291, (doi:10.1038/418291a).
- Lashuel, H. A. et al. Alpha-synuclein, especially the Parkinson's disease-associated mutants, forms pore-like annular and tubular protofibrils. *J Mol Biol* (2002), 322, 1089-1102, (doi:10.1016/s0022-2836(02)00735-0).

These new results provide further evidence on the proposed mechanism of action while it cross-validates: (i) the use of type B* as mimics of oligomers generated in a complex aggregation reaction; (ii) the information provided by our time-resolved fluorescence spectroscopy analysis, where we describe the interaction of PSM α 3 with α S type B* oligomers and fibrils at single-particle resolution.

If by "mode of action" the reviewer refers to "which specific residues are involved in the contacts". Our article demonstrates that such interaction is not sequence-specific neither for the peptide

(redesigned variants) nor for α S (conformational specificity). We propose that the interaction is driven by a defined spatial distribution of complementary biophysical properties at the molecules/assemblies' surfaces. Accordingly, it is not expected -nor it fits with our data- that a few residue-to-residue-specific interactions would drive the peptide-aggregate complex formation.

Our analysis indicates that the overall biophysical properties of the solvent exposed surfaces of the molecules is what governs their association, particularly, a complementary combination of hydrophobic and positively charged surfaces of the amphipathic helical peptides with the hydrophobic-next-to-negatively-charged surfaces in the α S aggregates (features present exclusively in type B* oligomers and fibrils). Besides, we believe that the data extracted using dcFCCS and single-molecule analysis is extremely valuable to shed light on such complex interaction, providing single-particle resolution. For instance, we were able to calculate both the peptide affinity and the mean number of peptide molecules bounds to each of the analyzed species, an information that is not accessible in most studies.

Generally, we think that the time-resolved fluorescence spectroscopy data provides information that addresses some of the concerns exposed here and in the following points since it informs on: (i) State of the oligomer preparation. (ii) Direct observation of the interaction between the different species and the peptides at single-particle resolution. (iii) Mechanistic information on peptide binding - provided by dual-color single-particle fluorescence spectroscopy and complemented by the peptide redesign strategy.

The authors present this work as a rational design of alpha-synuclein inhibitors, but then do not show how the design of the peptide was guided by the structural insights from our current understanding of alpha synuclein oligomers and their diversity. Instead, they simply say that they identified the PSM α 3 and that it fulfilled the desired criteria.

Based on the known features of the different α S species, as previously reported in multiple high-quality publications (as explained above), we rationalized a mode of interaction in which surface complementary rather than specific residue-to-residue contacts would drive the selective binding of peptides to prefibrillar toxic oligomers and fibrils. The rational biophysics-based selection of these structural properties is central to our work.

Of course, the validity of the hypothesis should be demonstrated with an example, and we decided to use a naturally occurring candidate (PSM α 3). We did not intend to design a peptide from scratch, and misleading sentences that might lead to this assumption have been deleted from the manuscript. Since our proposed mechanism is not sequence-specific, PSM α 3 is as good as any other possible peptide candidate (natural or synthetic) and constitutes a first proof-of-principle of the mode of action in our manuscript.

Then, we devoted a significant part of the manuscript to test our original hypothesis and confirm step by step the properties driving the interaction. Our work contains a number of non-natural

rational designs engineered to individualize these properties. Their successful deconvolution crystallized in the generation of a minimized 19-residues peptide scaffold with only 4 different residues that recapitulates the interacting properties and ultimately validates our original hypothesis and the initial selection of PSM α 3. In our opinion this effective low-complexity a-Syn aggregation inhibitor peptide constitutes a notable exercise of protein design.

The entire manuscript is focused on designing specific helical peptide inhibitors based on exploiting the structural properties of four particular types of alpha-synuclein oligomers (A/A* and B/B*) and fibrils that the authors claim to possess distinct conformation, morphology, size, hydrophobicity, and toxic properties. Yet, nowhere in the manuscript do the authors present data on the biochemical and biophysical homogeneity/heterogeneity of the oligomers and their conformational properties. Simply citing the previous publications, some of which do not provide the information mentioned, is not sufficient and does not allow one to properly assess the experiments and the interpretation of the results from the experiments performed by the authors.

The authors claim on several occasions that they investigated the interactions of the peptides with four major alpha synuclein species, implying that they are dealing with homogeneous preparations of alpha synuclein oligomers and fibrils. However, all the methods used by the others to generate oligomers are known to yield heterogeneous mixtures of oligomers rather than a specific type of oligomers. The methods used here, repeated concentration procedures, and centrifugation through 100 KDa, could also alter the structure and distribution of the oligomers in the samples.

Therefore, it is imperative that the authors provide the following data on their oligomer and fibril preparations.

- 1) Electron microscopy data (Multiple EM images) that allow for assessment of the morphology of the oligomers in each preparation.
- 2) Light scattering or sedimentation velocity to assess the size distribution of the oligomers.
- 3) Electron microscopy data on the fibril preparations.
- 4) SDS-PAGE analysis of each preparation to establish their purity and the absence of truncated alpha synuclein species.
- 5) Amounts of monomers in each preparation.

The design rationale is on the basis that the authors have obtained structural data on individual types of oligomers, which is not possible given the heterogeneity of the samples. They do not present any data to demonstrate that the oligomers they used possess the design features they used to target (hydrophobic patches embedded in the anionic environment) or present data that demonstrate how these features are different in different types of oligomers.

We truly appreciate the comments and questions raised by reviewer 3. As reported previously, the protocols to generate the different α S aggregated species yield remarkably reproducible preparations. However, in order to demonstrate the quality and properties of our samples without leaving room for doubts, we have characterized the size, morphology, purity, structure and hydrophobicity of the α S species studied in the manuscript by means of SDS and native PAGE electrophoresis, atomic force microscopy (AFM) analysis, dynamic light scattering (DLS), infrared (IR) spectroscopy and anilino-naphthalene-8-sulfonic acid (ANS) fluorescence spectroscopy; we have included these data in the supplementary information (**Supplementary Figure 1**).

The more detailed structural characterization of the oligomeric species performed by cryo-EM and solid-state NMR can be found in:

- Chen SW, et al. Structural characterization of toxic oligomers that are kinetically trapped during alpha-synuclein fibril formation. *PNAS* (2015). 112:E1994-E2003 (doi: 10.1073/pnas.1421204112)
- Fusco G, et al., Structural basis of membrane disruption and cellular toxicity by alpha-synuclein oligomers. *Science* (2017) 358:144-30 (doi: 10.1126/science.aan6160)

The reviewer can find a summary of the main findings at the end of the text in this document.

As for the data required by the reviewer, we think that such data is now present in the new Supplementary Figure 1, in figure 3, and in the Supplementary Figures 9 and 10. On the other hand, some of these concerns (i.e. size distribution of the oligomers) are inherently evaluated in the time-resolved fluorescence spectroscopy data. To further clarify the suitability of our sample, here we include an SDS-PAGE analysis of α S preparation before and after the aggregation reaction (**Figure R2**), showing the absence of truncated variants or dimers.

Figure R2: SDS-PAGE of α S aggregation reaction before (time 0 hours) and after (time 32 hours) of incubation. All samples were boiled for 5 minutes before application in order to denaturate fibrillar species.

The authors also indicate that the oligomers were kept at room temperature for three days. It is not clear why they did this, but what is important is to establish that the biophysical properties did not change during this incubation period. Usually, oligomer preparations are stored at -20C or 4C.

Data has been published that clearly show the disaggregation of oligomeric preparations at 4°C. In contrast, only a small amount of the preparation disaggregates at room temperature; the stability analysis can be found here: Chen SW *et al* and Cremades N. Structural characterization of toxic oligomers that are kinetically trapped during alpha-synuclein fibril formation. *PNAS* (2015). 112:E1994-E2003.

The same behavior has been reported for α S fibrillar samples: Ikenoue T *et al* and Goto Y. Cold denaturation of alpha-synuclein amyloid fibrils. *Angew Chem Int Ed Engl* (2014); Chen SW and Cremades N. Preparation of α -Synuclein Amyloid Assemblies for Toxicity Experiments. *Methods in Molecular Biology* (2018).

Although both oligomeric and fibrillar samples have been reported to be stable for more than a week at room temperature, we prefer to work within the first two-three days after preparation to be as conservative as possible, dcFCCS indicates that the preparations are stable. In any case, it is important to note that different monomer/oligomer ratios would not compromise the measures since we are working at single-particle resolution and not measuring ensemble averages.

The authors should present data on the oligomerization states of the peptides. They should establish using robust biophysical methods the oligomerization state of these peptides and whether they bind to alpha synuclein as monomers or oligomers.

We acknowledge that the reviewer's question is whether it is the monomeric or oligomeric form of the peptide that interacts with the amyloid species. As suggested by reviewer 1, the peptide may present a degree of oligomerization due to its amphipathic nature. This yields a diffusion coefficient of 19 $\mu\text{m}^2 \text{s}^{-1}$, for the peptide, which most likely corresponds to an equilibrium between monomeric and oligomerized species. As also noted by reviewer 1, the peptide used as a negative control (**Supplementary Figure 8**) shows a faster diffusion (108 $\mu\text{m}^2 \text{s}^{-1}$), because it lacks the amphipathic helix nature and, therefore, does not self-assemble.

Our binding curves in **Figure 2** and **Figure 6** support that monomeric peptide can bind to the aggregates because N_p values as low as 1 are observed for type A*, type B* oligomers and fibrils. The same is true for the binding curves and the N_p values obtained by single-particle stoichiometry analysis, as shown in **Supplementary Figure 7**. Besides, the N_p titration curves are

well fitted to one binding mode, indicating that most of the peptide, if not all, binds to the α S aggregates in its monomeric form.

This is an important point that is now more clearly explained in the revised version of the manuscript (page 9, lines 210-214) and further discussed in the supplementary information (lines page 6, 151-156).

The authors do not present any data that establish the mode of binding to different α -syn species and which regions of alpha synuclein interact with these peptides.

We understand the reviewer's concern. However, in the article, we characterize the interaction of different peptides with four isolated α S species at single-molecule resolution. To further provide mechanistic information on such interaction, we (i) Calculate the affinity of the peptide for each aggregate and the maximum number of peptide binding sites. (ii) Deconvolve the peptide properties responsible for the interaction and even achieve to obtain a simplified peptide version with no sequence diversity that concatenates such properties.

The mechanism that the data in the manuscript distillates is that the interaction is not governed by the specific interaction of well-defined α S residues uniquely suited to interact with particular peptide residues. As previously stated, it is the presence of some critical complementary structural properties that drive the interaction. Thus, we believe that defining the interaction in terms of complementary surfaces is an accurate and suitable descriptor of the interaction. Indeed, the demonstration of the binding of a re-designed simplified peptide version with no sequence diversity but with the right combination of surface properties to the α S prefibrillar oligomers and fibrils reinforces our proposal. Considering the complexity of oligomers and the lack of sequence specificity of the interaction, it may be extraordinarily challenging and off the scope of this manuscript, to characterize this interaction at the residue level.

The authors do not provide any data that establishes the exact mode of action for the peptide inhibitors or show that they indeed act by preventing oligomer to fibril transition, as they claim in the paper (page 9). They rely mainly on ThT signal intensity of EM, both of which are not quantitative. To show that these peptides indeed target oligomers and assess which species along the pathway of aggregation they target, they need to use more quantitative techniques to determine the distribution of the different alpha synuclein species in the presence and absence of peptide inhibitors. This can be achieved easily using simple sedimentation- or SEC-based techniques. If their proposed mechanisms of action are correct, then they should be able to see the accumulation of alpha synuclein oligomers-peptide complexes. Also, the EM data for all the peptides should be included as supporting information.

We agree with the reviewer and, as suggested, we have conducted new experiments to isolate the α S low-molecular weight species at different time points populating the aggregation reaction both in the absence and the presence of the peptide. To this end, we have applied a centrifugation-

based filtration protocol developed by Kumar and coworkers (Kumar, S. T et al., J Neurochem (2020), doi:10.1111/jnc.14955, see supplementary information) and the distribution of the isolated low-molecular weight species has been analyzed by SDS-PAGE and by TEM to assess the size and morphological diversity.

The SDS-PAGE analysis (**Figure R3**) captures quantitative changes in the concentration of low molecular weight α S oligomeric species along the aggregation pathway in the presence of the peptide compared to the untreated sample, being especially evident after 12h of aggregation. Importantly, the TEM analysis revealed that after 12 hours in control aggregation reactions α S mainly forms small fibrillar species and round prefibrillar aggregates (average diameter between 20 and 40 nm) (**Figure 3d** and **Supplementary Figure 10**). In contrast, at the same timepoint, samples incubated with the peptide contained a large fraction of small oligomers of annular shape with diameters between 9-14 nm, morphologically similar to the annular oligomers previously identified during in vitro α S aggregation and type B* oligomers (see the references at the beginning of Reviewer 3' response). Together with the dual-color time-resolved fluorescence spectroscopy data, this evidence strongly suggests that the peptide could be preventing or retarding the conversion of these annular oligomers - similar to type B* oligomers- into fibrillar species. Importantly, these results endorse the use of the kinetically stabilized type B* oligomers as mimics of on pathway oligomers in complex aggregation reactions.

Despite we understand the reviewer's suggestion on using alternative quantitative techniques (sedimentation or SEC), the low concentration of low-molecular weight aggregates in our assay conditions prevents any of these analyses. The aforementioned analysis would require working at much higher protein concentrations, thus changing the aggregation conditions in which we report the inhibition. We think that the SDS-PAGE and TEM (together with the time-resolved fluorescence spectroscopy) provides solid experimental evidences on the mechanism of inhibition and the species targeted along the aggregation reaction.

We truly believe that the new analysis we performed upon the reviewer's suggestion has significantly improved the quality and cohesion of the new submitted manuscript by bringing together the data obtained with the single-molecule time-resolved fluorescence spectroscopy analysis and the bulk aggregation kinetics.

Accordingly, a new paragraph with all these results has been added in the 'Results' section (Lines 249-261 in the main text), together with a new panel in **Figure 3**, a new supplementary figure (**Supplementary Figure 10**) and the description of the experimental process in the 'Methods' section in the supplementary information.

Figure R3: Analysis of low molecular weight species generated during the aggregation reaction. SDS-PAGE showing the distribution of low molecular weight aggregates along the aggregation reaction in the absence and presence of PSM α 3. TEM micrographs show the differential morphologies between the treated and the untreated sample at 12 hours.

We also provide below an additional figure with the requested EM data. Yet, in our opinion, it may not be necessary to generate a new supplementary figure in the manuscript with these data (**Figure R4**).

Figure R4: TEM micrographs of end point aggregation reactions in presence of equimolar concentrations of All_Leu (a), All_Leu19 (b), Scaffold_19 (c), Anionic_scaffold (d) and LL-37 (e).

It is puzzling that the authors use the same exact ThT data for aSyn in all the figures despite the fact that the various experiments were performed in different periods. One would expect that they included alpha synuclein alone samples in each aggregation experiments and the effect of the

inhibitors should be compared to the untreated alpha synuclein samples in the same experiment and in the same plate. They should explain and justify this.

We understand the reviewer concern. Indeed, aggregation reactions were all repeated when elaborating the manuscript. They were done for all the peptides simultaneously and in the same experiment in order to provide robust, reliable and comparable results.

The authors claim that they used the absorbance at 280 nm to determine the concentration of the oligomers. However, this method may not be accurate for the oligomers prepared by co-incubation of alpha-synuclein with EGCG.

The reviewer is right in his/her appreciation. Since type A* oligomers are only used in the dual-color single-particle fluorescence spectroscopy experiments where the concentration is directly calculated from the number of particles in solution extracted directly by dcFCCS, the exact concentration of the oligomers was adjusted for each preparation, in each individual measurement.

We have clarified this point in the revised version of the supplementary information (page 4, lines 83-85).

There is no point of comparing the activity of the peptide inhibitors here to other inhibitors unless they were assessed under the same conditions and in the same assays. Therefore, the comparison to SynuClean-D and other molecules (Anle138b) here is not valid or fair.

In fact, SynuClean-D is a compound developed by our research groups and assayed in **Figure 3a** simultaneously with our peptides using the same exact protocol. Thus, we think that it is a valid comparison. It is true, however, that we did not test Anle138b and accordingly any reference to this compound has been removed in the new version of the manuscript.

The ability of the peptide to abolish alpha synuclein-oligomers-induced increase ROS is not consistent with the fact that it only inhibited oligomer membrane binding and uptake by only ~60%.

From our perspective, a 60% reduction in membrane binding and oligomer uptake could well explain the ROS data. Although the level of induced ROS species correlates with the number of oligomers interacting with the cell, the correlation does not seem to be linear according to previously published research articles: Perni M et al., A natural product inhibits the initiation of alpha-synuclein aggregation and suppresses its toxicity. *PNAS* (2017) 114: E1009-17 (doi: 10.1073/pnas.1610586114); Perni M et al., Multisetp inhibition of alpha-synuclein aggregation and toxicity in vitro and in vivo by trodusquemine. *ACS Chem Biol.* (2018) 13: 2308-19 (doi: 10.1021/acscchembio.8b00466).

The authors suggest that the peptide LL-37 exhibit similar tissue distribution as alpha synuclein without showing any data to support their claim. They should elaborate on this and present the data.

We understand the reviewer's concern, as it is important to clarify the fact that it is well established in the bibliography that both α S and LL-37 are expressed in the brain and gastrointestinal tract. According to the reviewers suggestions, we have clarified this issue in the 'Results' and 'Discussion' sections restricting our asseveration to this specific tissues/organs (Lines 353-355 and 414-422, respectively) of the new submitted version of the manuscript.

LL-37:

-Lee, M. et al. Human antimicrobial peptide LL-37 induces glial-mediated neuroinflammation. *Biochemical Pharmacology* (2015).

-Burton, M.F. et al. The chemistry and biology of LL-37. *Natural Product Reports* (2009).

α S:

-Breen, D.P. et al. Gut-brain axis and the spread of α -synuclein pathology: Vagal highway or dead end? *Movement Disorders* (2019)

-Stolzenberg, E. et al. A Role for Neuronal Alpha-Synuclein in Gastrointestinal Immunity. *J Innate Immun* (2017).

Summary of the main structural features of Type B* and Type A* alpha-synuclein oligomers reported in previous publications (the figure numbers corresponds to those of the original publications: (Fusco G. et al., Cremades N, Ying L, Dobson CM, De Simone A. Structural basis of membrane disruption and cellular toxicity by alpha-synuclein oligomers. Science (2017) 358:1440; Chen SW et al., and Cremades N. Structural characterization of toxic oligomers that are kinetically trapped during alpha-synuclein fibril formation. PNAS USA(2015) 112: E1994)): [redacted]

Reviewers' comments:

Reviewer #1 (Remarks to the Author):

The authors have responded to all of the comments that I previously made, and I am mostly content with the changes made and the new version of the manuscript.

My only issue is with the authors' claims of specificity/selectivity. The authors claim to have addressed this issue with "more precise phrasing". I am still concerned that their claims could be misconstrued, as the specificity/selectivity is between the different α -syn species, and not the α -syn species compared with other proteins, or aggregates formed from other proteins. For example, the abstract still implies that they have a peptide that selectively binds α -syn oligomers, whereas what they actually have is a peptide that binds α -syn oligomers more selectively than α -syn monomer. They need to be clearer with their claims.

Reviewer #3 (Remarks to the Author):

I appreciate the author's efforts to address many of the points I raised in my review. However, many of the critical concerns have not been addressed.

Main issues:

Binding data: The authors did not perform any of the experiments requested to validate the specificity of the binding they extrapolated from their fluorescence measurements. It was suggested that they need to demonstrate the specificity of binding to oligomers or fibrils using other techniques (e.g., binding to Oligomer A and Oligomer B by SPR or the techniques)

Aggregation Assays: The authors failed to address previous comments about reliance on only ThT data to establish inhibition and mode of action of the peptide inhibitors. The point was made that a more quantitative assay that allows the assessment of differences in the relative amount of the three species (monomers, oligomers, and fibrils) in the presence and absence of inhibitors. These experiments, which are simple to conduct, were not done. Instead, the authors still rely mainly on EM data from a single time point to show oligomers' presence in samples containing the peptide inhibitor. Without the requested data, there is no direct evidence that these peptides block α Syn fibrillization by stabilizing intermediate oligomers and blocking their transition to fibrils. As suggested previously, if this is the case, one should observe a significant accumulation of oligomers or oligomers and monomers by SEC or other techniques. These experiments are simple and could be done in a couple of days.

They do not provide evidence for the isolation (by SEC) of oligomer-peptide complexes, which should be easy to do if these peptides bind as described.

In the new experiment they present in the rebuttal (Figure 3), the results argue against their claims. In the untreated sample, one would expect not to see any accumulation of oligomers as most of the sample converts to fibrils as suggested by ThT. In contrast, significant oligomer accumulation should be seen in the treated α Syn samples at 12 hours, instead one sees the opposite. Furthermore, the intensity of the band in the treated sample changes only slightly. The authors focus on the EM but do not address the differences in the distribution of the various species. This is why it is crucial to quantitatively assess the distribution of all three species, monomers, oligomers, and fibrils, and not to rely only on imaging techniques.

They show more accumulation of oligomers in the case of the untreated samples instead

Peptides oligomerization state: The author's data suggested that the peptide inhibitors have a propensity to self-associated, yet when asked to experimentally assess the oligomerization state of the peptide inhibitors, they argued that it is not necessary, despite that these experiments can be easily performed using techniques they have in the lab, e.g., light scattering and sedimentation velocity/equilibrium. They simply rely on the interpretation of their fluorescence data to suggest that the peptides exist in a monomeric state. This data is important to determine whether the monomeric or oligomeric form of these peptides binds to the aSyn oligomers.

Specificity:

1. The authors did not explain how these peptides bind to oligomers and fibrils. Given the major differences in structure and architecture of the two aggregation states, one would expect to see major differences in binding and surface properties. Today, we have access to large number of Cryo-EM structure of different types of aSyn fibril produced in vitro or isolated from MSA brains. it is not clear why they did not leverage this data to explain their results or how these peptides bind to the fibrils. Do these peptides bind to the different types of aSyn fibril structure or the brain-derived fibril structures? Do they bind to fibrils of other amyloid proteins?

2. The authors present this work as a rational design of alpha-synuclein inhibitors but then do not show how the design of the peptide was guided by the structural insights from our current understanding of aSyn oligomers and their diversity or how their data and conclusions are consistent with what we know about the structure of aSyn fibrils. They talk about surface complementary but do not show detailed analyses of the oligomers' surfaces and the fibrils.

3. In the discussion section, they suggest that these newly discovered peptides represent more powerful tools for the development of biomarker assays, compared to antibodies, mainly because they suggest that many aSyn antibodies bind to oligomers and monomers. However, in this manuscript, they never addressed the specificity of these peptides towards other amyloid oligomers or fibrils (Abeta or Tau or other amyloid proteins).

4. Membrane binding and Toxicity: The "toxicity experiments" lack the proper controls, aSyn monomer, oligomer A and fibrils. A direct comparison of all four species should be at performed at least once.

5. Toxicity: The assays used here are not toxicity assays as the authors do not link the increase in ROS to neither viability nor cellular dysfunction.

6. Characterization of oligomers: The authors have provided additional data on the oligomers' characterization, Type A non-toxic (EGCC-induced oligomers) and Type B toxic.

* The four species should've been characterized by SDS-PAGE and native gel. Instead, monomers and fibrils are analyzed by SDS-PAGE and oligomers by native gel, where they do not enter the gel. The behavior of these two types of oligomers in denaturing and non-denaturing gels is well documented. Under the conditions used, it is not possible to assess the purity of these samples or assess the amounts of monomers in these samples.

* The SDS page provided in the rebuttal raises some concerns. They show virtually identical intensity of the monomer band at 0 and 32 h and the absence of high molecular weight smear. This is unusual as boiling is not sufficient to disassociate oligomers to monomers. Usually, in such samples, one still see smearing above the monomer, which is completely absent in the gel they present here.

* The radius of oligomers B by light scattering does not match with the dimensions measured by AFM

* Despite differences in the biochemical, size, and structural properties the two types of oligomers, the author report in Figure 1 that they contain the same number of aSyn monomers (30).

A* oligomers, 4.4 ± 0.9 nm height and 32 ± 5 nm diameter

tB* oligomers, and a 6.3 ± 0.3 nm height, 95 ± 14 nm

* Note that the oligomers used in the fluorescence experiments were labeled and thus subjected to additional manipulations. Therefore it is important to establish that both their structural and binding properties are similar to the unmodified oligomers.

* The authors were asked to demonstrate the stability of oligomers at room temperatures. They did not provide the data and instead point to previous studies. The stability always has to be assessed under the exact conditions used here. They mainly discuss previous studies on the stability at 4C, but not under the conditions used here. At room temperature, they suggest that the oligomers are stable for weeks but do not show the data or cite previous data. Again, these experiments are simple, do not use a large amount of sample, and take only a couple of days.

*The term pathogenic species should be replaced by toxic oligomers as these oligomers' pathogenic properties were not assessed in vivo. We do not know to what extent they resemble oligomers in the brain.

* The absence of characterization of the oligomers in the presence of the peptide inhibitors makes challenging to determine if their binding to oligomers alters the morphological properties of the oligomers or their size distribution.

Reviewers' comments:

Reviewer #1 (Remarks to the Author):

The authors have responded to all of the comments that I previously made, and I am mostly content with the changes made and the new version of the manuscript.

My only issue is with the authors' claims of specificity/selectivity. The authors claim to have addressed this issue with "more precise phrasing". I am still concerned that their claims could be misconstrued, as the specificity/selectivity is between the different α -syn species, and not the α -syn species compared with other proteins, or aggregates formed from other proteins. For example, the abstract still implies that they have a peptide that selectively binds α -syn oligomers, whereas what they actually have is a peptide that binds α -syn oligomers more selectively than α -syn monomer. They need to be clearer with their claims.

We understand the reviewer's concern. We have carefully revised the manuscript, and all the sentences referring to "selectivity" or "specificity" have been rephrased as "conformational selectivity" or "conformational specificity". Considering this suggestion, we have changed the manuscript's title to: " α -helical peptidic scaffolds to target α S toxic species with nanomolar affinity", leaving the selectivity term aside.

Reviewer #3 (Remarks to the Author):

I appreciate the author's efforts to address many of the points I raised in my review. However, many of the critical concerns have not been addressed.

We genuinely appreciate the reviewer's dedication to our work. We understood the concerns raised in the previous revision, and thus we tried to clarify them in the preceding revision. We are very sorry to read that the reviewer is not satisfied with our efforts to address his/her points, which has led to the editor's rejection decision, despite the other two experts' favorable opinion.

We sincerely believe that all the critical concerns have already been answered, if technically possible, or justified adequately if not. In this sense, we want to stress that some of the requested experiments could not be done as he/she proposed because those techniques have critical limitations that preclude their implementation in our analysis. It is also frustrating to read some almost identical comments to those we answered in the previous revision without further discussion of our reply.

We firmly believe that we provided orthogonal evidence that clarified the relevant reviewer's concerns in the previous revision. In any case, we will try to make them more apparent in the following point-by-point response.

Main issues:

Binding data: The authors did not perform any of the experiments requested to validate the specificity of the binding they extrapolated from their fluorescence measurements. It was suggested that they need to demonstrate the specificity of binding to oligomers or fibrils using other techniques (e.g., binding to Oligomer A and Oligomer B by SPR or the techniques).

The time-resolved fluorescence spectroscopy analysis techniques employed in this article (both dcFCCS and single-molecule) are gold standard techniques in the field for the quantitative analysis of protein-protein interactions and of exceptional relevance for the characterization of complex systems in solution (Bacia and Schwille 2007), (Hellenkamp, et al. 2018), (Hillger, et al. 2007)). As stated in the first appeal, the data extracted using these techniques can not be accessed by alternative, in the bulk approaches. For instance, we were able to calculate both the peptide affinity and the mean number of peptide molecules bounds to each of the analyzed species in solution.

In fact, the here employed techniques are considered by the community as direct observations (Yang, et al. 2018), (Huang, et al. 2009), (Cremades, et al. 2012), (Shammas, et al. 2015), (Orte, et al. 2008)) due to their single-molecule resolution. In that sense, we think that some of the reviewer's comments distillate the underestimation of our time-resolved fluorescence spectroscopy analysis. In his/her review, he/she employs the terms "*extrapolated from their fluorescence measurements*" or "*They simply rely on the interpretation of their fluorescence data*". We fully disagree with this phrasing since we firmly believe that our binding analyses are of the highest quality possible. Indeed, reviewer 1, who, according to the editor, is an expert in "single-molecule methods and super-resolution microscopy" did not cast any doubt on the quality of our data.

We agree that SPR (which was not mentioned in the initial revision) could be an orthogonal approximation to study the interactions. Nevertheless, it has the major drawback of requiring surface immobilization, which can create artifacts and may not represent what is happening in the solution. Importantly, α S has been proved to be a difficult protein to work with in SPR. Indeed, few reported studies have been successful in applying SPR with monomeric α S and none, that we are aware of, with α S oligomers. Even if possible, SPR would not provide higher quality information than the here employed approaches, and in the best case, it will be confirmatory information. Therefore, the lack of an SPR analysis should not question the obtained results either hinder the potential consideration for acceptance of this article.

Aggregation Assays: The authors failed to address previous comments about reliance on only ThT data to establish inhibition and mode of action of the peptide inhibitors. The point was made that a more quantitative assay that allows the assessment of differences in the relative amount of the three species (monomers, oligomers, and fibrils) in the presence and absence of inhibitors. These experiments, which are simple to conduct, were not done. Instead, the authors still rely mainly on EM data from a single time point to show oligomers' presence in samples containing the peptide inhibitor. Without the requested data, there is no direct evidence that these peptides block aSyn fibrillization by stabilizing intermediate oligomers and blocking their transition to fibrils. As suggested previously, If this is the case, one should observe a significant accumulation of oligomers or oligomers and monomers by SEC or other techniques. These experiments are simple and could be done in a couple of days.

First, we would like to highlight that it has been experimentally demonstrated (for example (Buell, et al. 2010), (Xue, et al. 2017)) and it is well established and largely used in detailed kinetic analysis of amyloid inhibition (for example (Michaels, et al. 2020), (Linse, et al. 2020), (Arosio, et al. 2016)) that ThT fluorescence intensity correlates linearly with amyloid

concentration over a wide range of ThT concentrations, including those we have used in this study, once the appropriate controls have been carried out, as it is the case in our study. In addition, it is also well established that it is difficult to distinguish between oligomers and short fibrils by standard biophysical techniques based on the size of the species, so the techniques proposed by the reviewer are not as useful as he/she claims for the quantitative analysis of monomers, oligomers and fibrils.

We would like to have further data on the inhibition and mode of action in an aggregation reaction mixture. However, such a task is extremely challenging and far from being trivial, despite this seems to be the reviewer's opinion. Two main factors hinder such kind of analysis: (i) Oligomer abundance along an aggregation reaction is very low, <3% of total protein (Cremades, et al. 2012). (ii) Aggregation reactions are very heterogenous; monomers, transiently populated oligomers, and fibrils coexist at the same time in what is frequently called an "aggregation soup". Accordingly, quantifying the amounts of oligomers can only be achieved by applying very sophisticated strategies. Including other molecules in the analysis, which might also oligomerize, exponentially increases the level of complexity. Then, these experiments are not simple to conduct.

For these reasons, the use of isolated samples of oligomers (type B* and type A*) and their analysis by time-resolved fluorescence spectroscopy is central to this work since it allows the direct observation of peptide-assemblies interactions at single-particle resolution, avoiding the measurement of ensemble averages. Using bulk techniques in more complex and heterogeneous samples (during aggregation kinetics) is not appropriate for addressing these interactions' nature, especially in the early stages where there is a >95% excess of monomer. The reviewer's insistence on applying bulk techniques and the affirmation that it is easy to derive specificity parameters from them is, in our eyes, surprising.

Regarding the reviewer's hypothesis, he/she interprets that blocking oligomers implies a significant accumulation of such species that, then, can be easily quantified. Such assertion is misleading. From a kinetic point of view, we note that blocking just a small fraction of the oligomers formed in the assay's timeframe would have a substantial impact on the aggregation kinetics since it is a rate-limiting step (Cremades, et al. 2012). Thus, even if a certain accumulation of these species occurs, it only involves a small fraction of the reaction's total protein. This is illustrated in the **Figure R3** of the previous response to reviewers (**reproduced below, unmodified and now named figure R1**); only very faint bands can be appreciated in the low-molecular-weight fractions purified from the aggregation reaction at different time points, which is indeed what one would expect from a kinetic point of view.

In the untreated sample, a more intense band can be observed, but, as the EM data illustrates, it does not correspond to spherical oligomers but to low-molecular-weight fibrillar species that were not sedimented in the ultracentrifugation step. This is expected, as the first fibrillar species are necessarily small and cannot be efficiently segregated from the oligomers. This "contamination" exemplifies the complexity of isolating and quantifying these species in an ongoing aggregation reaction.

Hence, it is evident that oligomer quantification cannot be assessed by bulk techniques such SEC or sedimentation-based analysis as suggested by the reviewer, due to their low abundance and the inherent challenge of separating oligomers and low-molecular-weight

fibrils. In particular, we find extremely surprising the suggestion of an SEC-based analysis when it implies the use of a significantly higher concentration of protein. To provide an example, in a thoughtful work of the Lashuel lab (Kumar, et al. 2020) dedicated to the isolation of α S monomers, oligomer, and fibrils; from an initial sample at 12 mg/mL, which was aggregated in conditions favorable for oligomer formation, they only obtained an oligomer peak with a maximum of 14 absorbance units. In another setup Melki lab (Pieri, et al. 2016) used SEC to purify oligomers from an aggregation reaction, but this required 7 days at 4 °C at a concentration > 11 mg/ml. We work at 1 mg/mL, which is quite a standard for α S kinetic assays. Since protein aggregation is strongly concentration-dependent, increasing the concentration will dramatically change the aggregation kinetics, the species at equilibrium and would become useless for the understanding of the inhibition we see. Of course, changing the temperature will also change the kinetics, and although 4 °C can be instrumental for isolating certain oligomers, it is far from physiological conditions.

Overall, the experimental evidence we provided is neglected, and the requests he/she made seem to disregard the experimental conditions required to attain them and their low relevance for the subject of study. Our Th-T data is reliable in demonstrating inhibition of fibril formation, which EM confirms. Additionally, our morphological analysis of the low-molecular-weight species (**Figure R1**, **Figure 3** in the manuscript and **Supplementary Figure 10**) demonstrates that oligomers morphologically similar to type B* oligomers accumulate in the presence of the peptide. Even if such accumulation only corresponds to a small fraction of the total protein, it is kinetically expected (**Figure R1**). These data are in strong agreement with a thoughtful single-particle analysis that provides solid evidence on the proposed mechanism of action.

Figure R1: Analysis of low molecular weight species generated during the aggregation reaction. SDS-PAGE showing the distribution of low molecular weight aggregates along

the aggregation reaction in the absence and presence of PSM α 3. TEM micrographs show the differential morphologies between the treated and the untreated sample at 12 hours.

They do not provide evidence for the isolation (by SEC) of oligomer-peptide complexes, which should be easy to do if these peptides bind as described.

We are surprised by this suggestion. As previously discussed, SEC requires aggregation reactions with a concentration an order of magnitude above our working conditions. Additionally, we will not have any evidence if we are purifying oligomer-peptide complexes, oligomers, or small fibrillar species, since they will elute in the void volume, at least in our experimental setup. Our time-resolved fluorescence spectroscopy data provides enough evidence on the existence of peptide-oligomer complexes. We do not see how a hypothetical purification of such complex, without details on the stoichiometry or the binding constants, could increase the manuscript's quality.

In the new experiment they present in the rebuttal (Figure 3), the results argue against their claims. In the untreated sample, one would expect not to see any accumulation of oligomers as most of the sample converts to fibrils as suggested by ThT. In contrast, significant oligomer accumulation should be seen in the treated aSyn samples at 12 hours, instead one sees the opposite. Furthermore, the intensity of the band in the treated sample changes only slightly. The authors focus on the EM but do not address the differences in the distribution of the various species. This is why it is crucial to quantitatively assess the distribution of all three species, monomers, oligomers, and fibrils, and not to rely only on imaging techniques.

They show more accumulation of oligomers in the case of the untreated samples instead

This concern has been partially answered above when discussing this same figure (**Figure R1** of this response).

Here, the reviewer is mistaking low-molecular-weight aggregates for oligomers. As previously discussed, in the untreated sample, the band with higher intensity (12 hours) does not correspond to oligomers but to small fibrillar species (see **Figure R1**, **Figure 3** in the manuscript and **Supplementary Figure 10**) not sedimented in the ultracentrifugation.

This illustrates again the intrinsic complexity of the system and the impossibility to meet the reviewer's request; even if we are applying a very robust and well-characterized protocol, the isolation of these species, and interpretation of data in the course of an aggregation reaction is extremely challenging. Indeed, the morphological analysis by EM provides unequivocal evidence on the nature of these species.

Finally, the reviewer's comment "*the intensity of the band in the treated sample changes only slightly*" is correct and demonstrates that the oligomeric fraction is a minimum percentage of the total protein. This ultimately illustrates why the reviewer's requests cannot be addressed and that such suggestions were based on an inadequate interpretation of the species equilibrium during aggregation in the presence of inhibitory molecules.

Peptides oligomerization state: The author's data suggested that the peptide inhibitors have a propensity to self-associated, yet when asked to experimentally assess the

oligomerization state of the peptide inhibitors, they argued that it is not necessary, despite that these experiments can be easily performed using techniques they have in the lab, e.g., light scattering and sedimentation velocity/equilibrium. They simply rely on the interpretation of their fluorescence data to suggest that the peptides exist in a monomeric state. This data is important to determine whether the monomeric or oligomeric form of these peptides binds to the aSyn oligomers.

The oligomerization state of the peptide has already been assessed by time-resolved fluorescence spectroscopy. These techniques provide single-molecule experimental data on the oligomerization state, which in fact is not an interpretation but a direct observation. This is discussed in lines 207 of the manuscript and 134 and 151 of the methods section. Reviewer 1 considered this characterization good enough to clarify his/her concerns. Additionally, our time-resolved fluorescence spectroscopy data provide unequivocal evidence on the binding of individual molecules of the peptide to the α S aggregates, despite the fact that the peptide can exist in a oligomer state, as commented in the manuscript. We think it is pointless to repeat this same analysis using techniques with lower resolution such as sedimentation velocity/equilibrium, which indeed would not be able to resolve the state of the peptide that binds the α S aggregates.

Specificity:

1. The authors did not explain how these peptides bind to oligomers and fibrils. Given the major differences in structure and architecture of the two aggregation states, one would expect to see major differences in binding and surface properties. Today, we have access to large number of Cryo-EM structure of different types of aSyn fibril produced in vitro or isolated from MSA brains. it is not clear why they did not leverage this data to explain their results or how these peptides bind to the fibrils. Do these peptides bind to the different types of aSyn fibril structure or the brain-derived fibril structures? Do they bind to fibrils of other amyloid proteins?

Similar concerns regarding the mechanism of binding were already extensively discussed in the previous response to reviewers. We firmly believe that this information was already included in the manuscript. We will try to clarify this issue further here. Additionally, we append below the response to the reviewer for a similar question in the previous round (in italics) for the sake of clarity.

Briefly, in this article, we demonstrate that the peptide binding to oligomers and fibrils is determined by their surfaces' biophysical properties and is not a residue-to-residue interaction governed by sequence identity. As detailed in **Figure 1** of the manuscript and demonstrated in **Supplementary Figure 1**, these properties are shared by both oligomers and fibrils. Therefore, it is not expected that a defined interaction between residue "x" in the peptide and residue "y" in the aggregate would drive complex formation. Thus, the peptides' disposition and orientation in the surface of the aggregates is expected to display a certain degree of heterogeneity.

The suggestion that we model the interaction based on the available Cryo-EM structures of in vitro formed or in vivo extracted α S fibrils is surprising. We are sure that the reviewer is aware that the N- and, specifically, the C-terminal ends are disordered and not visible in these structures. The C-terminal region of α S is acidic and likely one of the regions

interacting with our peptides' cationic face. Therefore the lack of densities for these regions in the fibril structure precludes any modeling attempt.

The concerns about the peptide specificity towards other amyloid proteins are further addressed below in point 5.

Previous response to a similar question:

The authors do not present any data that establish the mode of binding to different α -syn species and which regions of alpha synuclein interact with these peptides.

We understand the reviewer's concern. However, in the article, we characterize the interaction of different peptides with four isolated α S species at single-molecule resolution. To further provide mechanistic information on such interaction, we (i) Calculate the affinity of the peptide for each aggregate and the maximum number of peptide binding sites. (ii) Deconvolve the peptide properties responsible for the interaction and even achieve to obtain a simplified peptide version with no sequence diversity that concatenates such properties.

The mechanism that the data in the manuscript distillates is that the interaction is not governed by the specific interaction of well-defined α S residues uniquely suited to interact with particular peptide residues. As previously stated, it is the presence of some critical complementary structural properties that drive the interaction. Thus, we believe that defining the interaction in terms of complementary surfaces is an accurate and suitable descriptor of the interaction. Indeed, the demonstration of the binding of a re-designed simplified peptide version with no sequence diversity but with the right combination of surface properties to the α S prefibrillar oligomers and fibrils reinforces our proposal. Considering the complexity of oligomers and the lack of sequence specificity of the interaction, it may be extraordinarily challenging and off the scope of this manuscript, to characterize this interaction at the residue level.

2. The authors present this work as a rational design of alpha-synuclein inhibitors but then do not show how the design of the peptide was guided by the structural insights from our current understanding of aSyn oligomers and their diversity or how their data and conclusions are consistent with what we know about the structure of aSyn fibrils. They talk about surface complementary but do not show detailed analyses of the oligomers' surfaces and the fibrils.

We tried to address these concerns in the previous revision, responding to an almost identical question. For greater clarity, we attach the previous response to the reviewer in italics at the end of the section. We infer that the reviewer is not satisfied with our previous answer, but it is not easy to provide a more concise discussion if he/she does not elaborate on our initial response.

Briefly, we based our design on previous analyses of both oligomers and fibrils. Type B* oligomers and fibrils expose hydrophobic patches to the solvent, while other α S oligomers do not (Fusco, et al. 2017), (Chen, et al. 2015) and are highly anionic. The anionic character of α S aggregates is evident, and the exposure of hydrophobic surfaces to the solvent is demonstrated in **Supplementary figure 1**, in good agreement with the literature (Fusco, et al. 2017), (Chen, et al. 2015). These two descriptors were sufficient to guide our design.

As already discussed, our design does not rely on a residue-to-residue interaction but the design of a surface complementary molecule (an amphipathic, cationic α -helical peptide). We understand that it may seem a simplistic approach, but it was accurate enough to initiate the project, and we ultimately demonstrated and validated each of the initial assumptions by rational design, as explained in the previous response.

About a detailed analysis of oligomer's surfaces; currently, there is limited information about the oligomers' three-dimensional structure (Fusco, et al. 2017), (Chen, et al. 2015). We expect that soon, a detailed description of the oligomers could lead to the structure-based design of ligands, but this is not possible nowadays. By contrast, several structures of α S amyloid fibrils could indeed be exploited to design new molecular entities. However, this information cannot be employed to improve our designs, since as detailed above, the structure of one of the principal contributors to binding, the C-terminal region, is not visible in the different polymorphs.

We applied a creative approach to deal with the limited structural information that we have on α S oligomers, from our perspective. The definitive evidence of the success of the design is that all the peptide variants generated work as intended.

The authors present this work as a rational design of alpha-synuclein inhibitors, but then do not show how the design of the peptide was guided by the structural insights from our current understanding of alpha synuclein oligomers and their diversity. Instead, they simply say that they identified the PSM α 3 and that it fulfilled the desired criteria.

Based on the known features of the different α S species, as previously reported in multiple high-quality publications (as explained above), we rationalized a mode of interaction in which surface complementary rather than specific residue-to-residue contacts would drive the selective binding of peptides to prefibrillar toxic oligomers and fibrils. The rational biophysics-based selection of these structural properties is central to our work.

Of course, the validity of the hypothesis should be demonstrated with an example, and we decided to use a naturally occurring candidate (PSM α 3). We did not intend to design a peptide from scratch, and misleading sentences that might lead to this assumption have been deleted from the manuscript. Since our proposed mechanism is not sequence-specific, PSM α 3 is as good as any other possible peptide candidate (natural or synthetic) and constitutes a first proof-of-principle of the mode of action in our manuscript.

Then, we devoted a significant part of the manuscript to test our original hypothesis and confirm step by step the properties driving the interaction. Our work contains a number of non-natural rational designs engineered to individualize these properties. Their successful deconvolution crystallized in the generation of a minimized 19-residues peptide scaffold with only 4 different residues that recapitulates the interacting properties and ultimately validates our original hypothesis and the initial selection of PSM α 3. In our opinion this effective low-complexity α -Syn aggregation inhibitor peptide constitutes a notable exercise of protein design.

3. In the discussion section, they suggest that these newly discovered peptides represent more powerful tools for the development of biomarker assays, compared to antibodies, mainly because they suggest that many α Syn antibodies bind to oligomers and monomers. However, in this manuscript, they never addressed the specificity of these peptides towards other amyloid oligomers or fibrils (A β or Tau or other amyloid proteins).

It is true that we did not experimentally address the specificity of these peptides to other amyloid proteins. However, in the discussion, we acknowledged that LL-37 inhibits the aggregation of A β 42 and IAPP, even if in those articles the main proposed mechanisms was an interaction with the monomer, mostly because these two peptides are inherently hydrophobic (GRAVY of 0.207 and 0.037 for A β 42 and IAPP, respectively). Instead, we show that LL-37 does not bind to monomeric α S and requires an oligomeric state of the protein with hydrophobic regions exposed to the solvent to interact; indeed, the GRAVY of monomeric α S is -0.403.

The described peptides or similar ones could bind to some degree other aggregates that satisfy the required properties; hydrophobic surfaces exposed to solvent and complementary electrostatic interactions. In those cases, the degree of complementarity will determine the interaction's affinity; that for α S aggregates is in the nanomolar range, clearly amongst the strongest described for a peptide- α S interaction. Even in the case that our peptides or derived ones would target the aggregates of other amyloids, this should not preclude its potential application in the clinics; a clear example is Anle138, a small molecule that acts as a generic oligomer modulator, binds to α S oligomers with much lower affinity than our peptides and is already in clinical trials (Wagner, et al. 2013).

From a biological perspective, the existence of human α -helical peptides that specifically bind α S aggregates encourages the analysis of such chaperone-like activity *in vivo*. Is it possible that such peptides play a relevant role in Parkinson's disease development? We are developing a mouse model to test this hypothesis. If this is eventually translated to other amyloid proteins, we could be in front of a hidden mechanism to prevent protein aggregation. However, exploring this hypothesis will be the subject of a future project, and it is not the aim of the here discussed manuscript.

Regarding their application in diagnosis platforms, we demonstrate that our peptides have nanomolar affinities to α S aggregates and do not bind to α S monomer even in a 500-fold excess, as monitored at single-particle resolution. To the best of our knowledge, there is no other molecule with those properties in the market. In fact, the implementation of the here described peptides for diagnosis has already been funded, and we are beginning to work on such a project with promising preliminary results.

4. Membrane binding and Toxicity: The "toxicity experiments" lack the proper controls, aSyn monomer, oligomer A and fibrils. A direct comparison of all four species should be at performed at least once.

The suggested controls would be suitable if we want to address type B* oligomers' toxicity and compare it with that of the other species. This is already well documented in publications in which we are directly involved (Fusco, et al. 2017), (Chen, et al. 2015). Since we want to test our peptides' detoxifying activity over type B* oligomers, α S monomer, oligomer A and fibrils are not the proper controls. Including the proposed comparison could be misleading and does not enrich the present article.

5. Toxicity: The assays used here are not toxicity assays as the authors do not link the increase in ROS to neither viability nor cellular dysfunction.

The reviewer's observation is correct; strictly talking, we did not use toxicity assays. However, type B* oligomers' toxicity via ROS is well-documented in previous works where we were directly involved (Deas, et al. 2016), (Angelova, et al. 2016), (Fusco, et al. 2017), and the increase in intracellular ROS is undoubtedly linked with cellular dysfunction and toxicity. Since we demonstrate that our peptides reduce oligomer's binding to cells and intracellular ROS levels, we think that is pertinent to state that our peptides have a detoxifying activity.

As we are sensitive to this discrepancy, and to clarify the issue, we changed the terms "oligomer toxicity" for "oligomer-induced damage" when referring to our cellular assays.

6. Characterization of oligomers: The authors have provided additional data on the oligomers' characterization, Type A non-toxic (EGCC-induced oligomers) and Type B toxic.

* The four species should've been characterized by SDS-PAGE and native gel. Instead, monomers and fibrils are analyzed by SDS-PAGE and oligomers by native gel, where they do not enter the gel. The behavior of these two types of oligomers in denaturing and non-denaturing gels is well documented. Under the conditions used, it is not possible to assess the purity of these samples or assess the amounts of monomers in these samples.

A native gel is appropriate to discriminate the amount of monomer in the oligomer samples since they migrate differently, as shown in **Figure 1B** of our previous publication in PNAS (Chen, et al. 2015). Additionally, as already discussed in the previous appeal, the monomer's relative concentration in the samples is also explicit in each time-resolved fluorescence spectroscopy measurement.

Either way, we provide below an SDS-PAGE of type B* oligomers as requested by the reviewer to assess their purity, although no additional information can be extracted.

Figure R2: SDS-PAGE of α S type B* oligomers. Samples were boiled for 5 minutes before application.

* The SDS page provided in the rebuttal raises some concerns. They show virtually identical intensity of the monomer band at 0 and 32 h and the absence of high molecular weight smear. This is unusual as boiling is not sufficient to disassociate oligomers to monomers. Usually, in such samples, one still sees smearing above the monomer, which is completely absent in the gel they present here.

We find it unfair that the reviewer casts doubts on the integrity of our experiments. We are surprised that once again, he/she directly raises concerns about the authenticity of our results and how despite our repeated efforts to demonstrate the quality of the provided

data, they are totally disregarded. We want to clarify that we have done an SDS-PAGE to assess the sample purity and that we have faithfully presented the results obtained in our standard working conditions.

For the sake of clarity, we attach the mentioned figure below (now renamed **Figure R3**), where no other species of molecular weight lower than 30 kDa are observed either, as expected:

Figure R3: SDS-PAGE of α S aggregation reaction before (time 0 hours) and after (time 32 hours) of incubation. All samples were boiled for 5 minutes before application.

* The radius of oligomers B by light scattering does not match with the dimensions measured by AFM.

Given the fact that the reviewer has misread the dimensions calculated by AFM in the following point, we assume that these discrepancies stem from this same mistake.

The proper dimensions derived from the AFM are:

Type A* oligomers: 5.1 ± 0.4 nm height and 28 ± 6 nm diameter.

Type B* oligomers: 4.4 ± 0.9 nm height and 32 ± 5 nm diameter.

Then the dimensions obtained by both techniques match in general terms and concur in a similar radius. It is important to note that slight deviations could be expected because:

(i) Oligomers are non-spherical particles. The hydrodynamic diameter obtained by DLS corresponds to the diameter of a sphere that has the same translational properties as the measured particle. Then in non-spherical particles, this may slightly differ from reality.

(ii) DLS measures the hydrodynamic radius, which may differ from the real particle dimension as it refers to how a particle diffuses within a fluid.

* Despite differences in the biochemical, size, and structural properties the two types of oligomers, the author report in Figure 1 that they contain the same number of aSyn monomers (30).

A* oligomers, 4.4 ± 0.9 nm height and 32 ± 5 nm diameter

B* oligomers, and a 6.3 ± 0.3 nm height, 95 ± 14 nm

The reviewer has misinterpreted the dimensions reported in the legend of **Supplementary Figure 1**. Such fragment is reproduced below in *italics* and tabulated to facilitate its reading:

a) AFM analysis of monomeric α S (top left), type A (top right), and type B* (bottom left) oligomers and sonicated fibrils (bottom 7 right) are shown. Statistical size distribution analysis yielded a -5.1 ± 0.4 nm height and 28 ± 6 nm diameter for type A* oligomers, -4.4 ± 0.9 nm height and 32 ± 5 nm diameter for type B* oligomers, and a -6.3 ± 0.3 nm height, 95 ± 14 nm width, and 300 ± 140 nm length for sonicated fibrils.*

Then, the dimensions stated by the reviewer are not those reported in the article. As it can be observed, the measured dimensions correspond to particles of very similar size and in agreement with our data and previous reports (Chen, et al. 2015), (Fusco, et al. 2017). Either way, it is important to clarify that, in addition to the number of monomers, type A* and type B* oligomers have different structural rearrangements and different degrees of disorder and compactness that can affect their particle size.

We also want to highlight that type A* and type B* oligomers display a distribution of sizes with slightly different numbers of monomers per oligomer as previously assessed by analytical ultracentrifugation (Fusco, et al. 2017). Then, values in **Figure 1** were an initial estimation based on previous analysis to illustrate the design and should not be taken as an absolute value for each particular oligomer. To prevent any other misinterpretation, we have substituted the number of monomers per oligomer in **Figure 1** for the exact average calculated *in situ* in our single-molecule analysis.

* Note that the oligomers used in the fluorescence experiments were labeled and thus subjected to additional manipulations. Therefore it is important to establish that both their structural and binding properties are similar to the unmodified oligomers.

As described in the methods section, oligomers were prepared using already labeled monomeric α S and were not subjected to additional manipulations after assembly as it is suggested. We acknowledge that the requirement of a label is a limitation of the technique that could slightly affect the properties of oligomers. However, it is important to note that labeling was explicitly designed to minimize such interference. We targeted position 122, a residue located in the C-terminal domain that is not part of the oligomer core and remains disordered. Labeled oligomers present identical characteristics than not labeled ones in terms of size and diffusion as assessed by FCCS (**Supplementary figure 1 and Figure 2**). Additionally, in a previous article, we performed a structural and morphological characterization of labeled and unlabeled oligomers by CD and AFM, and no significant differences were observed (see **supplementary figure 7** of the referenced article (Chen, et al. 2015)).

In any case, the presence of a label in the oligomers does not compromise the most critical points of our article, and thus, in our view, it should not question the potential consideration for acceptance of this article.

* The authors were asked to demonstrate the stability of oligomers at room temperatures. They did not provide the data and instead point to previous studies. The stability always has to be assessed under the exact conditions used here. They mainly discuss previous studies on the stability at 4°C, but not under the conditions used here. At room temperature, they suggest that the oligomers are stable for weeks but do not show the data or cite previous data. Again, these experiments are simple, do not use a large amount of sample, and take only a couple of days.

We do not understand why the reviewer insists on challenging the quality of our methodology or preparations that have been largely characterized and reported previously in the most prestigious scientific journals (PNAS, Science). We stick to well-established procedures in the field based on already published analyses performed by ourselves (Fusco, et al. 2017), (Chen, et al. 2015), (Chen and Cremades 2018). Additionally, as replied in the previous response to reviewers, our time-resolved fluorescence spectroscopy measurements inherently validate the oligomer preparation state as we directly monitor the stability of the different oligomers and fibrils (we detect both oligomers and monomers in the single-particle analysis)

In the first revision, the reviewer indicated that oligomers are usually stored at 4 °C or -20 °C. We replied that several published data demonstrate that oligomers are more stable at 20 °C, including recent reports where their stability at 4 °C and 20 °C is evaluated (Chen, et al. 2015), (Chen and Cremades 2018). As described in the methods section, we employed oligomers within the first two-three days after preparation as previously done (Fusco, et al. 2017), 25855634 (Chen, et al. 2015), 29953201 (Perni, et al. 2018). We firmly believe that bibliography sets a strong precedent that should not be overlooked, especially when we were involved in such publications.

Even if the reviewer chooses to neglect these precedents, we have already answered that the single-particle analysis employed in the article confirms the samples' stability. Each single time-resolved fluorescence spectroscopy measurement cross-validates the oligomer stability and would have allowed us to detect any reduction of the number of oligomers or changes in their diffusion/size.

*The term pathogenic species should be replaced by toxic oligomers as these oligomers' pathogenic properties were not assessed in vivo. We do not know to what extent they resemble oligomers in the brain.

Although the toxicity of these oligomers have been indeed assessed in vivo, in healthy mice upon injection into the brain and were observed to be toxic (Froula, et al. 2019), we agree with the reviewer that we still do not know if these oligomers are the pathogenic species involved in the disease. We have, therefore, modified the new version n of the manuscript, and the title of the manuscript accordingly: " α -helical peptidic scaffolds to target α S toxic species with nanomolar affinity".

* The absence of characterization of the oligomers in the presence of the peptide inhibitors makes challenging to determine if their binding to oligomers alters the morphological properties of the oligomers or their size distribution.

This is a good point. Our current time-resolved fluorescence spectroscopy data demonstrates that the interaction does not result in large changes in the oligomers size distribution since the diffusion of oligomers with peptide is fairly similar to that of oligomers alone (Cross-correlation curve in **Figure 2c** and **Figure 6e**). This suggests that the peptide interacts with oligomers but does not induce significant changes (disaggregation or further aggregation).

What is clear is that the peptide interaction changes the oligomer particles' surface properties, as shown by our cell-based assays.

We speculate that the peptide interaction results in a blockage of the exposed hydrophobic patches in the oligomer. Unfortunately, we could not obtain structural resolution on how the peptide bind to oligomers, nor on its disposition on their surfaces. Limited structural information on oligomers structure is available, and only Cryo-EM and ssNMR analysis have provided data of sufficient quality to approach oligomer morphological properties. Additionally, the here exposed mechanism of interaction is not sequence-specific, and several peptides bind to one oligomer, probably with different orientations, which complicates a high-resolution structural characterization.

To overcome these limitations, we are currently engaged in a collaboration with Prof. José María Valpuesta (CNB-CSIC, Madrid) to study such interaction using cryo-EM. It should be noted that what we expect to obtain are density maps without residue-resolution, similarly to what has been obtained for oligomers in previous characterizations also carried by Prof. Valpuesta. We are highly committed to following up on this project, and these cryo-EM experiments represent a significant step forward. In that context, we do not expect to have these results before a year from now, and if the resolution is good enough, we think they will merit an article on their own.

References

1. Bacia, K. & Schwille, P. Practical guidelines for dual-color fluorescence cross-correlation spectroscopy. *Nat Protoc* 2, 2842-56 (2007).10.1038/nprot.2007.410. <https://www.ncbi.nlm.nih.gov/pubmed/18007619>
2. Hellenkamp, B. et al. Precision and accuracy of single-molecule FRET measurements-a multi-laboratory benchmark study. *Nat Methods* 15, 669-676 (2018).10.1038/s41592-018-0085-0. <https://www.ncbi.nlm.nih.gov/pubmed/30171252>
3. Hillger, F., Nettels, D., Dorsch, S. & Schuler, B. Detection and analysis of protein aggregation with confocal single molecule fluorescence spectroscopy. *J Fluoresc* 17, 759-65(2007).10.1007/s10895-007-0187-z. <https://www.ncbi.nlm.nih.gov/pubmed/17447125>
4. Yang, J. et al. Direct Observation of Oligomerization by Single Molecule Fluorescence Reveals a Multistep Aggregation Mechanism for the Yeast Prion Protein Ure2. *J Am Chem Soc* 140, 2493-2503 (2018).10.1021/jacs.7b10439. <https://www.ncbi.nlm.nih.gov/pubmed/29357227>
5. Huang, F., Ying, L. & Fersht, A.R. Direct observation of barrier-limited folding of BBL by single-molecule fluorescence resonance energy transfer. *Proc Natl Acad Sci U S A* 106, 16239-44 (2009).10.1073/pnas.0909126106. <https://www.ncbi.nlm.nih.gov/pubmed/19805287>

6. Cremades, N. et al. Direct observation of the interconversion of normal and toxic forms of alpha-synuclein. *Cell* 149, 1048-59 (2012).10.1016/j.cell.2012.03.037. <https://www.ncbi.nlm.nih.gov/pubmed/22632969>
7. Shammas, S.L. et al. A mechanistic model of tau amyloid aggregation based on direct observation of oligomers. *Nat Commun* 6, 7025 (2015).10.1038/ncomms8025. <https://www.ncbi.nlm.nih.gov/pubmed/25926130>
8. Orte, A. et al. Direct characterization of amyloidogenic oligomers by single-molecule fluorescence. *Proc Natl Acad Sci U S A* 105, 14424-9 (2008).10.1073/pnas.0803086105. <https://www.ncbi.nlm.nih.gov/pubmed/18796612>
9. Buell, A.K., Dobson, C.M., Knowles, T.P. & Welland, M.E. Interactions between amyloidophilic dyes and their relevance to studies of amyloid inhibitors. *Biophys J* 99, 3492-7 (2010).10.1016/j.bpj.2010.08.074. <https://www.ncbi.nlm.nih.gov/pubmed/21081099>
10. Xue, C., Lin, T.Y., Chang, D. & Guo, Z. Thioflavin T as an amyloid dye: fibril quantification, optimal concentration and effect on aggregation. *R Soc Open Sci* 4, 160696 (2017).10.1098/rsos.160696. <https://www.ncbi.nlm.nih.gov/pubmed/28280572>
11. Michaels, T.C.T. et al. Thermodynamic and kinetic design principles for amyloid-aggregation inhibitors. *Proc Natl Acad Sci U S A* 117, 24251-24257 (2020).10.1073/pnas.2006684117. <https://www.ncbi.nlm.nih.gov/pubmed/32929030>
12. Linse, S. et al. Kinetic fingerprints differentiate the mechanisms of action of anti-Abeta antibodies. *Nat Struct Mol Biol* 27, 1125-1133 (2020).10.1038/s41594-020-0505-6. <https://www.ncbi.nlm.nih.gov/pubmed/32989305>
13. Arosio, P. et al. Kinetic analysis reveals the diversity of microscopic mechanisms through which molecular chaperones suppress amyloid formation. *Nat Commun* 7, 10948 (2016).10.1038/ncomms10948. <https://www.ncbi.nlm.nih.gov/pubmed/27009901>
14. Kumar, S.T., Donzelli, S., Chiki, A., Syed, M.M.K. & Lashuel, H.A. A simple, versatile and robust centrifugation-based filtration protocol for the isolation and quantification of alpha-synuclein monomers, oligomers and fibrils: Towards improving experimental reproducibility in alpha-synuclein research. *J Neurochem* 153, 103-119 (2020).10.1111/jnc.14955. <https://www.ncbi.nlm.nih.gov/pubmed/31925956>
15. Pieri, L., Madiona, K. & Melki, R. Structural and functional properties of prefibrillar alpha-synuclein oligomers. *Sci Rep* 6, 24526 (2016).10.1038/srep24526. <https://www.ncbi.nlm.nih.gov/pubmed/27075649>
16. Fusco, G. et al. Structural basis of membrane disruption and cellular toxicity by alpha-synuclein oligomers. *Science* 358, 1440-1443 (2017).10.1126/science.aan6160. <https://www.ncbi.nlm.nih.gov/pubmed/29242346>
17. Chen, S.W. et al. Structural characterization of toxic oligomers that are kinetically trapped during alpha-synuclein fibril formation. *Proc Natl Acad Sci U S A* 112, E1994-2003(2015).10.1073/pnas.1421204112. <https://www.ncbi.nlm.nih.gov/pubmed/25855634>
18. Wagner, J. et al. Anle138b: a novel oligomer modulator for disease-modifying therapy of neurodegenerative diseases such as prion and Parkinson's disease. *Acta Neuropathol* 125, 795-813 (2013).10.1007/s00401-013-1114-9. <https://www.ncbi.nlm.nih.gov/pubmed/23604588>
19. Deas, E. et al. Alpha-Synuclein Oligomers Interact with Metal Ions to Induce Oxidative Stress and Neuronal Death in Parkinson's Disease. *Antioxid Redox*

- Signal 24, 376-91 (2016).10.1089/ars.2015.6343.
<https://www.ncbi.nlm.nih.gov/pubmed/26564470>
20. Angelova, P.R. et al. Ca²⁺ is a key factor in alpha-synuclein-induced neurotoxicity. *J Cell Sci* 129, 1792-801 (2016).10.1242/jcs.180737.
<https://www.ncbi.nlm.nih.gov/pubmed/26989132>
21. Chen, S.W. & Cremades, N. Preparation of alpha-Synuclein Amyloid Assemblies for Toxicity Experiments. *Methods Mol Biol* 1779, 45-60 (2018).10.1007/978-1-4939-7816-8_4. <https://www.ncbi.nlm.nih.gov/pubmed/29886526>
22. Perni, M. et al. Multistep Inhibition of alpha-Synuclein Aggregation and Toxicity in Vitro and in Vivo by Trodusquemine. *ACS Chem Biol* 13, 2308-2319 (2018).10.1021/acscchembio.8b00466.
<https://www.ncbi.nlm.nih.gov/pubmed/29953201>
23. Froula, J.M. et al. Defining alpha-synuclein species responsible for Parkinson's disease phenotypes in mice. *J Biol Chem* 294, 10392-10406 (2019).10.1074/jbc.RA119.007743. <https://www.ncbi.nlm.nih.gov/pubmed/31142553>

Reviewers' comments:

Reviewer #1 (Remarks to the Author):

The authors' modified manuscript now makes it clear that the peptides are conformation-specific, and not protein-specific. My comments have been responded to, and I can therefore recommend the manuscript for publication.

Reviewer #3 (Remarks to the Author):

The authors were asked to provide data that validate their single-molecule results. They have chosen not to do this for any of the points raised in my last review.

1. Binding data: The authors did not perform any of the experiments requested to validate the specificity of the binding they extrapolated from their fluorescence measurements. Their claim that "Indeed, few reported studies have been successful in applying SPR with monomeric α S and none, that we are aware of, with α S oligomers" is not accurate and a simple search of the literature (aSyn and SPR) would show that this is not the case. Here is a partial list

Antibodies (SYNO2, SYNO3, and SYNO4) vs aSyn fibrils

<https://pubmed.ncbi.nlm.nih.gov/25937088/>

Nanobody vs aSyn variants (monomers)

<https://pubmed.ncbi.nlm.nih.gov/29603451/>

aSyn(A53T) vs single-domain intrabodies (Data is in Sinfo)

3) <https://pubmed.ncbi.nlm.nih.gov/30514850/>

ASyM or ASyO2 or ASyO5 vs monomers (Data is in Sinfo)

4) <https://www.ncbi.nlm.nih.gov/pmc/articles/PMC3949727/>

2. Regarding the ThT data, there are many reports in the literature where changes in the intensity or ThT kinetics do not correlate with the extent of aggregation as determined by independent protein-based quantitative methods. A recent paper published in Science Advances also reported on one type of aSyn fibrils that exhibits very low ThT binding (stealth fibrils) compared to other aSyn fibrils at the same concentration. These observations and others show that ThT alone is not a reliable quantitative method for establishing inhibition or mode of action of aggregation inhibitors.

A recent report on the discovery of ThT stealth aSyn fibrils

1) <https://advances.sciencemag.org/content/6/40/eabc4364>

ThT kinetics do not accurately report on the difference on the aggregation of WT and nitrated aSyn as determined by sedimentation assays (Figure 3)

<https://pubmed.ncbi.nlm.nih.gov/25768729/>

Differences on the ThT fluorescence intensities between aSyn variants

3) <https://www.tandfonline.com/doi/full/10.1080/13506129.2018.1517736>

4) <https://journals.plos.org/plosone/article?id=10.1371/journal.pone.0067713>

3. I agree that it could be challenging to distinguish between small fibrils and oligomers, but this depends on the extent of accumulation of each species and the size distribution of the fibrils, which was not assessed here.

4. Capturing aSyn oligomer: It is true that under spontaneous aggregation conditions only a small number of oligomers accumulate. However, previous studies have shown the accumulation of a

significant number of oligomers that could be separated by SEC in the presence of inhibitors. In fact, this is now a commonly used method to generate and isolate oligomers (e.g., dopamine, HEN, and EGCG). The authors claim that their peptides block fibrillization by stabilizing intermediate oligomers and blocking their transition to fibrils. Now, they are claiming that this is not possible to validate this experimentally because the amounts of oligomers in the sample is small. This would suggest that in the presence of peptide inhibitors, the great majority of the protein remains monomeric. What they were asked here is to simply verify this using an independent technique. Why would the monomer not form fibrils if it does not bind to the peptide or convert to oligomers? The question asked was simple, in the presence of these inhibitors, what is the distribution of aSyn monomers, fibrils, and oligomers? Kumar et al. recently reported simple sedimentation-based protocols that allow the separation of as little as 5% of oligomers from fibril samples. Thus, the proposed experiments are doable and remain important for understanding these peptide inhibitors' mode of action.

5. The authors were asked to verify the peptide's oligomerization state by other techniques, even for the most potent inhibitors. They argued that this is not necessary.

6. When asked to explore the possibility of modeling the interaction between the peptides and the fibrils' surfaces, they argued that it could not be done or that this exercise would not be informative. In the paper, they present models of the surfaces of their peptides. Cryo-EM structures of aSyn fibrils provide information about fibril surfaces. Therefore, it is not clear why this information cannot be used to explore the surface biophysical properties of the fibrils and to determine if their peptide inhibitors would be able or not to capture the diversity of the various fibril surfaces. The authors argue that the significant parts of the N- and C-terminal domains of aSyn are not visible in the Cryo-EM structure, which would preclude modeling the interactions based on structural information. This assumes that these regions play an important role in defining the interactions with the peptide inhibitors, which the authors have not shown. When asked previously about the sequence determinants of these interactions, they argued that it is not important.

7. Low molecular weight oligomers vs. oligomers: It is not possible to distinguish between low molecular weight oligomers and oligomers on the basis of the data they provide. Without systematic analysis of the distribution of oligomers detected here, it is not possible to make such a distinction, especially given the heterogeneity of the oligomer preparations. They report that low molecular weight species show spherical and ring-like structures that have previously been shown to exhibit molecular size in the range of 150-300 KDa. In the amyloid field, particularly abeta field, the term low molecular weight oligomers commonly refer to dynamic oligomers that are not visible by imaging techniques, dimer, trimer, tetramer, hexamer.

Reviewer #4 (Remarks to the Author):

This is an interesting and generally well written manuscript that describes the design and validation of a peptide-based inhibitor of alpha-synuclein aggregation that importantly targets only the toxic-species in the aggregation cascade. It uses a nice mixture of single / quasi-single molecule fluorescence methods and biochemical and cell biology assays to make their case.

I read the manuscript in full prior to reading the reviewer/response to reviewer comments.

Interestingly, I had four main concerns (the first one brought up by both referees and the second and third one by ref 2).

1. Despite the authors attempts at rectifying this by the use of the term "conformational selectivity" (whose meaning is unclear), there is no evidence in this manuscript that their peptide is specific for alpha-synuclein this means that its use as therapeutic /diagnostic is unclear.

2. As the ability to inhibit aggregation is central to this paper the authors really need to use another assay to validate this. For example, a monomer quantification assay using a spin down, followed by SDS-PAGE at the end point would be an orthogonal assay to show inhibition of fibril formation.
3. The degree of rational design or generation of a structure activity relationship is somewhat overstated. As discussed by the authors in their rebuttal, the C-terminal region (acidic) is not part of the structured core of the available fibril structures, so the relative pattern or demarcation of hydrophobic/-vely charged surface "seen" by the peptide is unclear (and a bit misleadingly schematically shown in Figure 1) and well beyond the precise positioning of the side-chains of an amphipathic helix
4. The cross-correlation and single molecule fluorescence methods to show that the peptide is interacting with specific alpha-syn species was very clear and nicely done. My only question on this was why the FRET efficiency changed so much between the oligomeric and fibrillar species?

Points 1, 3 and 4 could be addressed editorially by inclusion of suitable caveats and explanations.

Reviewer #4 (Remarks to the Author):

This is an interesting and generally well written manuscript that describes the design and validation of a peptide-based inhibitor of α S aggregation that importantly targets only the toxic species in the aggregation cascade. It uses a nice mixture of single / quasi-single molecule fluorescence methods and biochemical and cell biology assays to make their case.

Authors: We thank the reviewer for his/her positive comments on our work.

I read the manuscript in full prior to reading the reviewer/response to reviewer comments. Interestingly, I had four main concerns (the first one brought up by both referees and the second and third one by ref 2).

Points 1, 3 and 4 could be addressed editorially by inclusion of suitable caveats and explanations.

1. Despite the authors attempts at rectifying this by the use of the term "conformational selectivity" (whose meaning is unclear), there is no evidence in this manuscript that their peptide is specific for α S this means that its use as therapeutic /diagnostic is unclear.

Authors: The reviewer's observation is pertinent, and we can understand his/her concern. We address this point below:

First, the term "conformational selectivity" has been removed from the abstract and the main text and we have rephrased those sentences containing it to clarify that the peptides have not been proven to be specific for α S but that they can discriminate between α S native and toxic species; thus, they are α S species-specific. In our opinion, this selectivity is remarkable since all α S species share the same sequence. The binding of LL-37 to A β 42 and IAPP was already mentioned and is now briefly described in the discussion. We assume that with these text modifications, the reader will notice that the peptides' selectivity towards α S aggregated species does not imply that they are necessarily specific for the α S protein.

In any case, it is important to note that in contrast to the case of α S, for A β 42, the binding of LL-37 is driven by a certain degree of sequence similarity and cannot discriminate between A β 42 species. The binding of LL-37 to A β 42 upon its resuspension in PBS exhibits a $K_D = 13.3 \mu\text{M}$ and a $K_D = 20.3 \mu\text{M}$ when the same A β 42 sample is incubated for 24 days (De Lorenzi et al., J Alzheimers Dis 2017, 59(4):1213-1226). Thus, in both cases the interactions are much weaker than the ones we observe for the binding of LL-37 to α S toxic oligomers ($K_D = 3.62 \text{ nM}$) and fibrils ($K_D = 5.6 \text{ nM}$). In IAPP, the binding of LL-37 responds again to sequence similarity (Armiento et al., Angew Chem Int Ed Engl 2020, 59(31):12837), and LL-37 binds both monomers ($K_D = 88 \text{ nM}$) and fibrils ($K_D =$ not determined); binding to oligomers is also possible but was not characterized. Thus, the ability to target only toxic oligomers and fibrils without binding the functional monomeric form seems to be unique

for α S and in excellent agreement with the initial biophysical assumptions that guided the work. A short sentence describing these differences has been now included in the discussion.

We agree with the reviewer that the direct use of these peptides for therapy cannot be taken for granted and should be explored in detail in the future. In the revised version of the manuscript, we have tuned down the therapeutic assumptions, also sentences like “Scaffold _19 constitutes a representative of a generic family of peptides with potential therapeutic applications.” or “From a therapeutic perspective, this is a significant advantage, as they are not expected to interfere with the physiological functions of the soluble protein.” have been removed.

Still, it is our view that even if our peptides or derived ones turn to target the oligomers of other amyloids, this would not necessarily preclude their potential application in the clinic. In this sense, they would resemble Anle138b; a small molecule thought to act as a generic oligomer interactor. It binds to α S fibrils with a significantly lower affinity than our peptides (190 ± 120 nM) (the affinity for oligomers is not available), and is already in clinical trials for the synucleinopathies. Also, there is high interest in the field for using molecular chaperones for therapy, which also have a broad specificity for misfolded proteins and aggregates.

Of course, we are aware that the exogenous administration of these molecules is not a trivial task. For this reason, one of the most promising and exciting results of our work is the identification of an endogenous peptide, LL-37, with the ability to bind α S toxic species. The expression of the gene encoding for LL-37 is inducible by different molecules, like vitamin D or butyrate, that have already shown a protective effect against Parkinson’s disease. Whether there is a connection between the uptake of these molecules, the overexpression of LL-37, their binding to α S toxic species in vivo, and Parkinson's disease symptoms amelioration is the subject of an ongoing project.

Regarding the potential use of the peptides for diagnostic, the principal benefit is their ability to discriminate monomeric from toxic aggregated α S, which in a diagnosis platform would reduce the background noise caused by the excess of monomeric α S in biofluids.

Again, we have tuned down the diagnostic assumptions along the manuscript, i.e. in the abstract we have specified “... promising tools to assist diagnosis by discriminating between native and toxic α -synuclein species.” and in the discussion we have deleted the sentence “...make turn these molecules powerful tools for the implementation of diagnostic strategies in which they may act as nanosensors of α S species in biological fluids”

Of course, we can face off-target effects; still, we expect the use of these molecules to be more specific than the quantification of Th-T positive aggregates, which has been proposed as a method to follow disease progression (Horrocks et al., ACS Chem Neurosci 2016, 7(3):399-406). Indeed, we believe that a diagnosis platform based on an anti- α S antibody and our ligands would allow quantifying α S toxic aggregates with high sensitivity, while avoiding the background caused by the

excess of monomeric α S in the biofluid, and a funded project is ongoing in such direction. In the revised manuscript, we briefly discuss this potential implementation.

2. As the ability to inhibit aggregation is central to this paper the authors really need to use another assay to validate this. For example, a monomer quantification assay using a spin down, followed by SDS-PAGE at the end point would be an orthogonal assay to show inhibition of fibril formation.

Authors: We fully agree with the reviewer. As suggested, we conducted new experiments to address the inhibition of fibril formation orthogonally. We added a new supplementary figure (**Supplementary Figure 9**) that includes the suggested sedimentation assay that confirms fibril inhibition.

The results of the sedimentation assay suggested by the reviewer (**Supplementary Figure 9b**) are in excellent agreement with the recorded Thioflavin-T signals (spectra of the same samples are shown in **Supplementary Figure 9a**), thus confirming the inhibitory capacity of the peptides (both PSM α 3 and LL-37) with a Thioflavin-T independent assay. In both cases, treatment with the peptides significantly increased the amount of α S that remains soluble at the end of the experiment, relative to an untreated sample.

Supplementary Figure 9. Orthogonal validation of PSM α 3 and LL-37 anti-aggregational activity. (a) Thioflavin-T fluorescence spectra in the presence of monomeric α S (Soluble) and end-point α S aggregation reactions (70 μ M) performed in the absence (Untreated) and in the presence of 35 μ M of PSM α 3 or LL-37. Spectra were recorded from 460 to 600 nm with an excitation wavelength of 445 nm. (b) Characterization of the amount of soluble α S in end-point aggregation reaction samples after sedimentation. The same samples were analyzed by SDS-PAGE (top panel) and quantification was performed by measuring the absorbance at 280 nm ($\epsilon = 5960 \text{ M}^{-1} \text{ cm}^{-1}$). Protein quantities were measured in triplicate. Error bars represent the standard deviation (SD).

3. The degree of rational design or generation of a structure activity relationship is somewhat overstated. As discussed by the authors in their rebuttal, the C-terminal region (acidic) is not part of the structured core of the available fibril structures, so the relative pattern or demarcation of hydrophobic/-vely charged surface “seen” by the peptide is unclear (and a bit misleadingly

schematically shown in Figure 1) and well beyond the precise positioning of the side-chains of an amphipathic helix.

Authors: We understand the reviewer's concern.

We have substituted the expression “With a structure-function relationship in hand, we identified...” in the abstract by “A structure-guided search allowed identifying...” which reflects better the procedure we followed. In addition, the sentence “This defined SAR, usually absent or difficult to dissect in small molecules, should help in the development and diversification of novel candidates with increased activities employing protein engineering.” in the discussion has been changed to “This defined binding mode should help in the development and diversification of novel ligands with increased activities.”

Apart from these text changes we will try to clarify our rationale here:

When hypothesizing a molecular entity able to bind α S toxic species, we thought of a complementary molecule in terms of hydrophobic and electrostatic interactions. This molecule must contain a hydrophobic surface with the potential to interact with the oligomers/fibrils exposed hydrophobic patches. Additionally, we considered that nonpolar interactions may not be sufficient to achieve high-affinity binding and that electrostatic contributions may increase the avidity, while favoring the binding to polymeric forms of α S (due to their higher net charge). Therefore, we sought to include a positively charged region in our design able to interact with oligomers/fibrils solvent-exposed and highly anionic C-terminal tails (the properties of the C-terminal α S region are now better described in the manuscript). According to our view and as discussed in the manuscript, an amphipathic, cationic α -helical peptide would provide a structurally stable scaffold to merge both features.

Despite speculative, we present a schematic model that illustrates how we envision the peptide binding mechanism to α S toxic species (see below, for review purposes only). The binding surfaces for peptide interaction would consist of hydrophobic patches in the oligomers/fibrils surfaces and the anionic C-terminal tail of α S, which remains solvent-exposed and disordered in both α S oligomers and fibrils (Fusco et al., *Science* 2017, 15;358(6369):1440-1443., Li et al., *Nat Commun* 2018, 9(1):3609.). The peptide hydrophobic face would interact with hydrophobic patches in the fibril's surface, and nearby C-terminal tails (not necessarily from the same monomer) would contribute to binding by interacting with the cationic face of the peptide that faces the solvent upon the first interaction. We are working to provide structural data on this interaction by Cryo-EM with Prof. J.M. Valpuesta (Centro Nacional de Biotecnología, Madrid, Spain) and expect to confirm or disregard this view in the future.

Figure R1. Schematic representation of the oligomer-peptide binding mechanism proposed model. In the design, the peptide randomly moves until it finds an exposed hydrophobic patch at the oligomer's surface to bind. The peptide's cationic surface is now oriented in a way that it faces the outside of the oligomer and, due to its electrostatic complementarity, can interact and attract the flexible exposed and highly negatively C-terminal α S region. Hence, the peptide interaction results in a blockage of the oligomer assembly capacity by hiding the hydrophobic patches.

A difference between the above shown oligomer scheme and the one we provide in Figure 1 is that we include here the C-terminal tails. Still, because the tails are not structurally visible, we preferred not to depict them in the main document Figure, while now describing the reason for that in the legend.

The foreseen mechanism of binding is in line with the experimental evidence reported in the article. According to the hypothesized model, binding affinity would arise from two different energetic contributions: 1) the apolar surfaces exposed hydrophobic patches of the oligomer, and the hydrophobic surface of the peptide fit together to exclude water molecules from the interface, resulting in favorable entropy; and 2) the polar surfaces, highly negative charge α S C-terminal tails (15 E/D) and the peptide cationic region make electrostatic interactions, resulting in favorable enthalpy.

As described in the text, we rationally re-designed a set of peptide variants to provide indirect support for the binding mode.

- PSM α 3 and LL-37 (cationic (+2 and +6) and amphipathic α -helical peptides): present a selective interaction with α S toxic species. The dcFCCS and single-particle fluorescence spectroscopy data determined a low nanomolar affinity, specific for type B-like oligomers and fibrils.

The fact that the peptides do not bind the monomeric α S evidences that the central NAC region's hydrophobicity is not sufficient to promote the peptide interaction.

- K9P-F11P PSM α 3 peptide: the PSM α 3 α -helical fold is disrupted. No binding with any of the α S species is detected. Thus, the α -helical conformation and the associated amphipathic character, with an asymmetric distribution of charge and hydrophobicity, seems to be a requirement for the interaction and the anti-aggregational properties.

- Scaffold-19 (cationic (+2) and amphipathic α -helical peptide): a simplified peptide version with no sequence diversity at the hydrophobic face but with the right combination of surface properties. This peptide presents the same anti-aggregational properties as PSM α 3 and LL-37. Thus, the interaction is not driven by sequence-specific residue interactions, allowing to describe the interaction in terms of complementary surfaces.
- Anionic-Scaffold (anionic (-2) and amphipathic α -helical peptide): This amphipathic, but positively charged peptide does not inhibit α S amyloid aggregation. Thus, the cationic character at the hydrophilic face is a requirement for interaction and anti-aggregational activity, which argues for a contribution of the anionic α S tail to the peptide oligomer/fibril interaction.

In summary, the obtained experimental data agree with the way we envision the α S toxic species/peptide interactions.

4. The cross-correlation and single molecule fluorescence methods to show that the peptide is interacting with specific alpha-syn species was very clear and nicely done. My only question on this was why the FRET efficiency changed so much between the oligomeric and fibrillar species?

The reviewer has spotted an interesting difference in the average FRET efficiency values of the donor and acceptor dyes in the peptide: α S complexes between the oligomeric and fibrillar species of alpha-syn. We believe this is due to the different arrangement of the α S molecules in the analysed oligomers with respect to the fibrils. Concretely, the fibrils present a parallel arrangement of the α S molecules, while this configuration is antiparallel in the oligomers (Chen et al., PNAS 2015, 112(16):E1994-2003.). This different inter-molecular distribution will originate different average distances between donor and acceptor fluorophores in the peptide: α S complexes, which will be reflected in different average FRET efficiencies.

In any case, we just used the FRET efficiency analysis to demonstrate further complex formation between the peptides and the α S species, as only in a complex can these molecules be in close distance to promote FRET between the fluorophores. Due to the complex analysis of FRET experiments with multiple donors and acceptors in proximity (in the complexes, more than one donor and one acceptor will be in FRET-competent distances), we precluded further analysis, and we just used it to prove complex formation by the appearance of significant FRET efficiencies.

REVIEWERS' COMMENTS

Reviewer #4 (Remarks to the Author):

The authors have provided a detailed explanation and made suitable amendments to their manuscript to address all of my concerns.

Whilst it would have been nice to see the monomer in solution assay for the dPSMalpha3 control for completeness it is not required. A second minor point is that the provided SDS-PAGE gel image is cropped.

Reviewer #4 (Remarks to the Author):

The authors have provided a detailed explanation and made suitable amendments to their manuscript to address all of my concerns.

Authors: We thank the reviewer for his/her positive opinion on the new experiments and manuscript changes.

Whilst it would have been nice to see the monomer in solution assay for the dPSMalpha3 control for completeness it is not required. A second minor point is that the provided SDS-PAGE gel image is cropped.

Authors: We share the opinion that dPSMalpha3 was not strictly required in this particular experiment.

An uncropped image of the SDS-PAGE data shown in Supplementary Figure 9 is included in the Source Data file. In any case, we show it also below.